# Long-term isolation of European steppe outposts boosts the biome's conservation value

Philipp Kirschner [1,8] ✉, Eliška Záveská[2,8], Alexander Gamisch[1,3], Andreas Hilpold[4], Emiliano Trucchi[5], Ovidiu Paun [6], Isabel Sanmartín [7], Birgit C. Schlick-Steiner[1], Božo Frajman[2], Wolfgang Arthofer[1], The STEPPE Consortium*, Florian M. Steiner[1,9] ✉ & Peter Schönswetter[2,9] ✉

The European steppes and their biota have been hypothesized to be either young remnants of the Pleistocene steppe belt or, alternatively, to represent relicts of long-term persisting populations; both scenarios directly bear on nature conservation priorities. Here, we evaluate the conservation value of threatened disjunct steppic grassland habitats in Europe in the context of the Eurasian steppe biome. We use genomic data and ecological niche modelling to assess pre-defined, biome-specific criteria for three plant and three arthropod species. We show that the evolutionary history of Eurasian steppe biota is strikingly congruent across species. The biota of European steppe outposts were long-term isolated from the Asian steppes, and European steppes emerged as disproportionally conservation relevant, harbouring regionally endemic genetic lineages, large genetic diversity, and a mosaic of stable refugia. We emphasize that conserving what is left of Europe's steppes is crucial for conserving the biological diversity of the entire Eurasian steppe biome.

---

[1] Department of Ecology, University of Innsbruck, Technikerstraße 25, 6020 Innsbruck, Austria. [2] Department of Botany, University of Innsbruck, Sternwartestraße 15, 6020 Innsbruck, Austria. [3] Department of Biosciences, University of Salzburg, Hellbrunnerstrasse 34, 5020 Salzburg, Austria. [4] Institute for Alpine Environment, Eurac Research, Drususallee 1/Viale Druso 1, 39100 Bozen/Bolzano, Italy. [5] Department of Life and Environmental Sciences, Marche Polytechnic University, Via Brecce Bianche, 60131 Ancona, Italy. [6] Department of Botany and Biodiversity Research, University of Vienna, Rennweg 14, 1030 Vienna, Austria. [7] Real Jardín Botánico CSIC, Plaza de Murillo 2, 28014 Madrid, Spain. [8]These authors contributed equally: Philipp Kirschner, Eliška Záveská. [9]These authors jointly supervised this work: Florian M. Steiner, Peter Schönswetter. *A list of authors and their affiliations appears at the end of the paper. ✉email: philipp.kirschner@gmail.com; florian.m.steiner@uibk.ac.at; peter.schoenswetter@uibk.ac.at

The Eurasian steppes are the second-largest continuous biome on Earth covering up to 7% of the Earth's total land surface[1] and play a major ecological role, for example as a global carbon sink[2]. Bordered by boreal forests to the North and (semi-)deserts to the South, the 10.5 million square kilometres of steppes range, in their potential natural extent without interruption, from the Pontic plains in Europe to China (Fig. 1a, Supplementary Fig. 1). By hosting up to 72 herbaceous plant species per 10 square metres, they even exceed the diversity of tropical rainforests at this spatial scale[3,4]. Eurasian steppes are also one of the most threatened biomes worldwide[5,6], mostly due to agricultural intensification[1,5,7], which has already led to a loss of about one million square kilometres of steppic grassland on the territory of the former USSR[8].

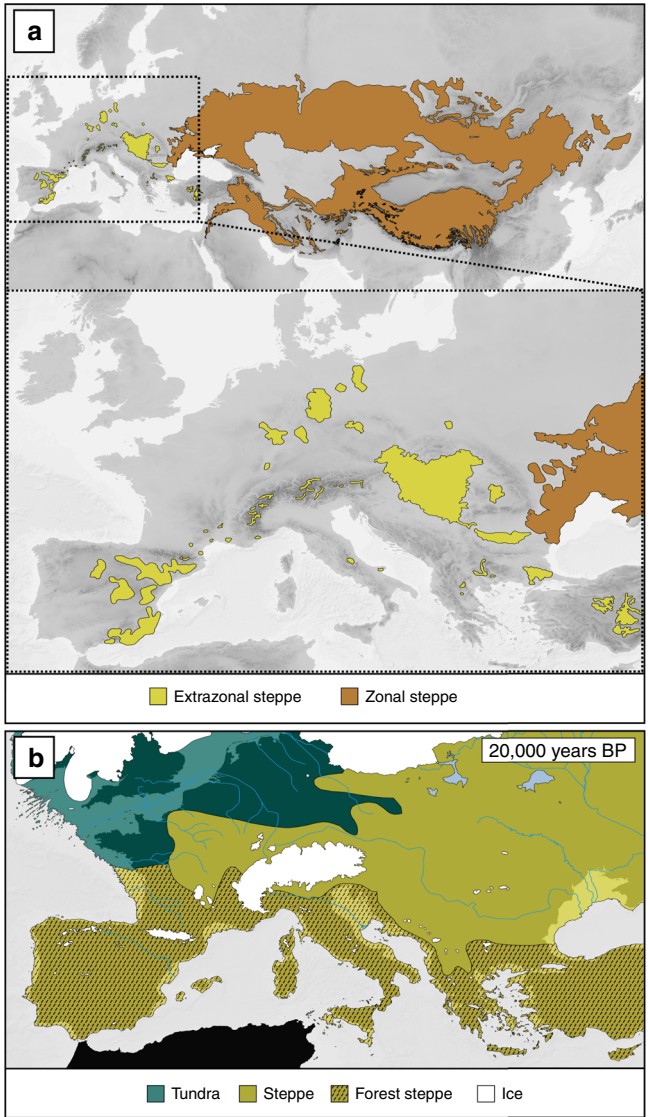

**Fig. 1 The Eurasian steppe biome. a** "Zonal steppe" denotes areas where the macroclimatically determined potential vegetation is steppes; their extent is based on Wesche et al.[2]. "Extrazonal steppe" highlights areas where steppe outposts occur embedded in a matrix of forest vegetation (for details on delimitation of extrazonal steppe extent see Supplementary Methods). Shades of grey depict topography. **b** Distribution of major vegetation types[71], extent of ice cover[62] (white), and position of coastline[62] (outline of vegetation layers) in western Eurasia at the Last Glacial Maximum, 20,000 years before present (BP). The underlying map depicts present-day Europe. Source data are provided as a Source Data file.

Outside the continuous distribution of the zonal steppes, which is primarily determined by macroclimate, outposts of steppic grasslands, that is, extrazonal steppes, occur wherever local climatic, topographic, and edaphic factors provide conditions resembling the zonal ones. In Europe, these highly disjunct islands of extrazonal steppes are dispersed within broadleaf forests and restricted to relatively dry, continental areas spanning the Carpathians in the East to the Iberian Peninsula in the West (Fig. 1a)[9]. The potential extent of these extrazonal steppes under natural conditions, that is, before large-scale anthropogenic landscape transformation, is unknown. Considering that the potential maximum extent of the largest extrazonal steppes in Europe in the Pannonian basin is 37,000 km², it is likely that the total extent of extrazonal steppes never exceeded 1% of the area of the zonal steppes[10].

The largely overlapping species composition of extrazonal and zonal steppes has served as the basis for the hypothesis[11–13] that the European extrazonal steppes and their biota are relatively young remnants of the zonal steppe belt that covered large parts of Eurasia during cold stages of the Pleistocene[11,12] (Fig. 1b). According to this hypothesis, shallow or no phylogenetic divergence among zonal and extrazonal populations is expected. Conversely, extrazonal steppe biota instead could have persisted past warm interglacials in refugia that overlap with their present-day occurrences. Under this scenario, evolutionary processes such as genetic drift and/or ecological adaptation would have caused deep divergence among zonal and extrazonal lineages[14] (the terms zonal lineage and extrazonal lineage refer to the source area of the respective lineage hereafter). In the latter case, the conservation value of extrazonal steppe biota would increase considerably and render them far more important units of biodiversity conservation than previously assumed. We emphasise that resolving these two contrasting hypotheses could substantially change the current perspective on the conservation of extrazonal steppes.

Currently, 1.4 million km² of zonal steppes are situated within protected areas, which is a considerable portion even though not equally distributed over the entire steppe belt[1]. In contrast, extrazonal steppic grasslands protected under the European Union's habitat directive cover only 6841 km²[15], which is negligible in terms of absolute size compared with the protected zonal steppes. Considering this imbalance along with the fact that zonal and extrazonal steppes share many species, one could legitimately question whether it is necessary to invest resources for the protection and management of the extrazonal European steppes.

We apply a complementary approach, combining genomic data, phylogenetic and population genetic inference, as well as ecological modelling across six steppe species representing different phyla, families, and genera sampled across the Eurasian steppe biome. In particular, we compare climatic niche, phylogenetic history, and evolutionary potential[16] of three flowering plants, which are widespread and abundant elements of the Eurasian steppes (the legume *Astragalus onobrychis*, the spurge *Euphorbia seguieriana*, and the grass *Stipa capillata*) and three arthropods (the grasshoppers *Omocestus petraeus* and *Stenobothrus nigromaculatus*, and the ant *Plagiolepis taurica*) to address two nested research questions. First, in prioritising among different parts of the Eurasian steppes, do we need to consider the isolated islands of extrazonal steppes in Europe as conservation relevant, or do they genetically represent just an outpost of the zonal Eurasian steppe belt? To evaluate this question, we define six criteria to assign conservation value to the extrazonal versus zonal areas (Fig. 2a–f). Specifically, we evaluate species by species (A) the geographic position of the deepest phylogenetic splits inferred from reduced-representation genome data; (B) the patterns of admixture between extrazonal and zonal populations

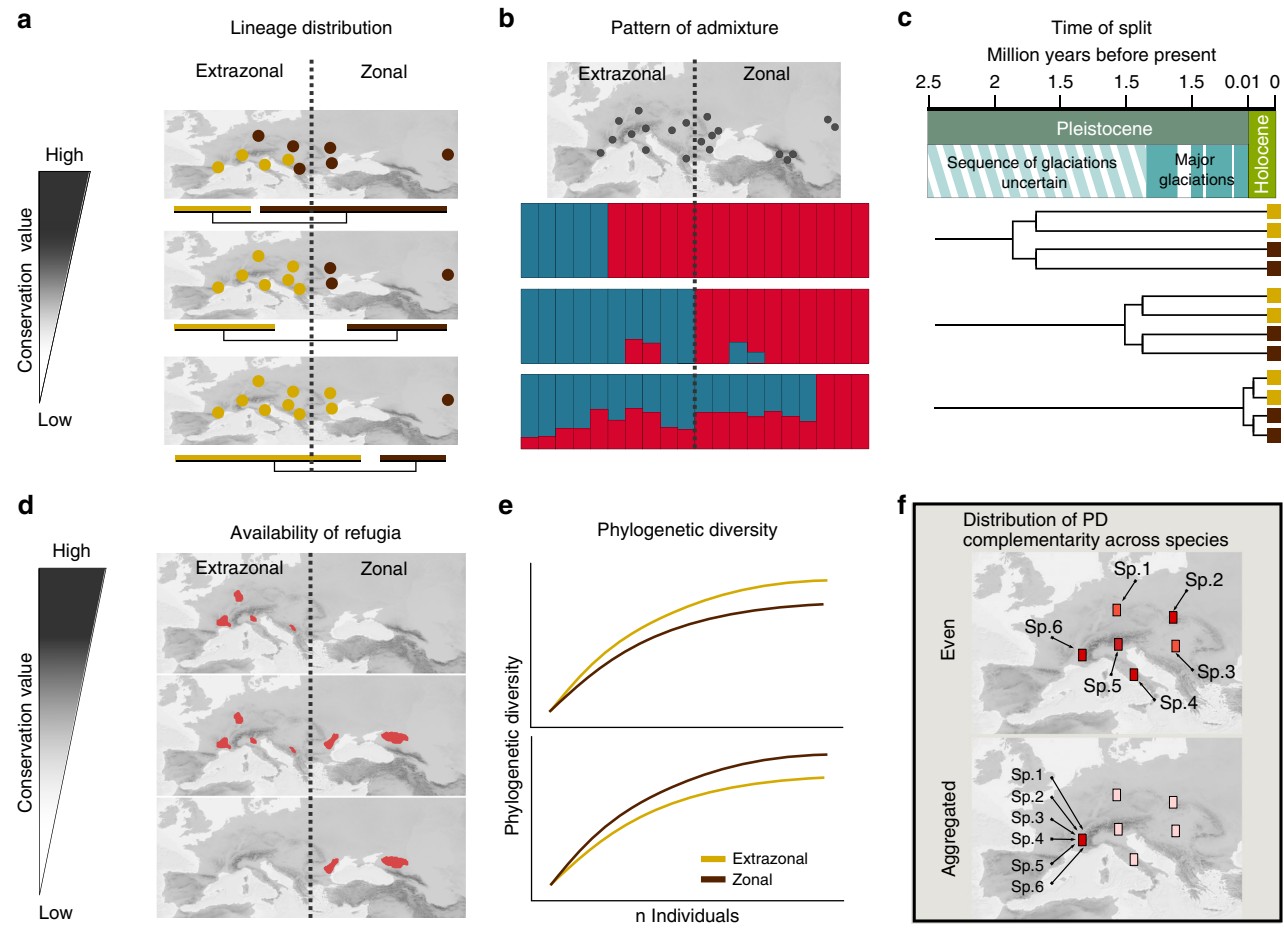

**Fig. 2 Criteria chosen to evaluate the conservation value of biota of European extrazonal steppes. a** Distribution of lineages throughout zonal and extrazonal steppes; populations illustrated as circles, colours represent origin of populations (yellow: extrazonal lineage, brown; zonal origin). **b** Geographic distribution of observed gene pools and admixture between them. **c** Time of lineage divergence (colour scheme as in (**a**)). **d** Geographic position of refugia. **e** Phylogenetic diversity (yellow line: extrazonal, brown line: zonal). **f** Distribution of phylogenetic diversity complementarity (PD complementarity) across species (Sp.1–6) in the extrazonal steppes: even distribution of PD complementarity across species versus spatial aggregation of PD complementarity across all species in a single area (dark red: large value; light red: small value); the scale indicating conservation value does not apply in this figure panel.

using Bayesian clustering based on the same data; (C) the ages of the main splits between the observed lineages of the animals using dated mitochondrial phylogenies; (D) the availability of suitable habitats (i.e. refugia) throughout Pleistocene cold and warm stages using ecological niche models (ENMs); (E) the phylogenetic diversity (PD) of the zonal versus the extrazonal steppes. Additionally, we ask if—in case the extrazonal steppes were conservation relevant—conservation efforts should focus on the extrazonal steppes as a whole or just on particular parts. To address this question, we employ the concept of PD complementarity to assess the extent of unique PD of the extrazonal steppes as a whole and of individual parts of the extrazonal steppes compared with the zonal steppes[17] (Fig. 2f). Based on the criteria defined, we demonstrate that the extrazonal steppes have large conservation value. We show that biota of extrazonal steppes were long-term isolated from the zonal steppes and are independently evolving entities. At the same time, extrazonal steppes also harbour lineages that immigrated from the zonal steppes. Consequently, conservation of extrazonal European steppes is key to conserve the Eurasian steppe biome as a whole.

## Results

**Genomic data reveal divergence of extrazonal populations.**
For 1036 accessions (Supplementary Data 1) of six characteristic steppe species (three angiosperms, three arthropods) sampled across their distribution ranges, we produced genome-wide single nucleotide polymorphisms (SNPs) extracted from restriction-site-associated DNA sequencing data (RADseq; Supplementary Table 1).

Phylogenetic inference exhibited well-supported clades that consisted of exclusively extrazonal lineages in all species (Fig. 3). Three general patterns emerged. In one species (*E. seguieriana*), extrazonal lineages were clearly nested in and thus derived from zonal lineages (pattern i)[18]. For all other species, extrazonal lineages were either (ii) restricted to parts of the European Alps and the Italian Peninsula (*S. capillata*, *S. nigromaculatus*) or iii) inhabited the Alps and western and southern Europe and met zonal lineages in or at the periphery of the Pannonian basin (*A. onobrychis*, *O. petraeus*, *P. taurica*). With the exception of *E. seguieriana*, these extrazonal lineages were always separated from zonal lineages by the deepest observed splits. These deepest splits were well-supported (except in *S. capillata*), while support within the subclades, albeit geography-correlated, differed across species and will not be further considered here.

Bayesian clustering revealed an optimal separation into two clusters for all species (Fig. 3, Supplementary Fig. 2). In four out of six species, these genetic clusters were completely coherent with the two main phylogenetic lineages (i.e. extrazonal and zonal lineages). In these four species, little or no admixture between these clusters was observed. In the case of *S. capillata*, the

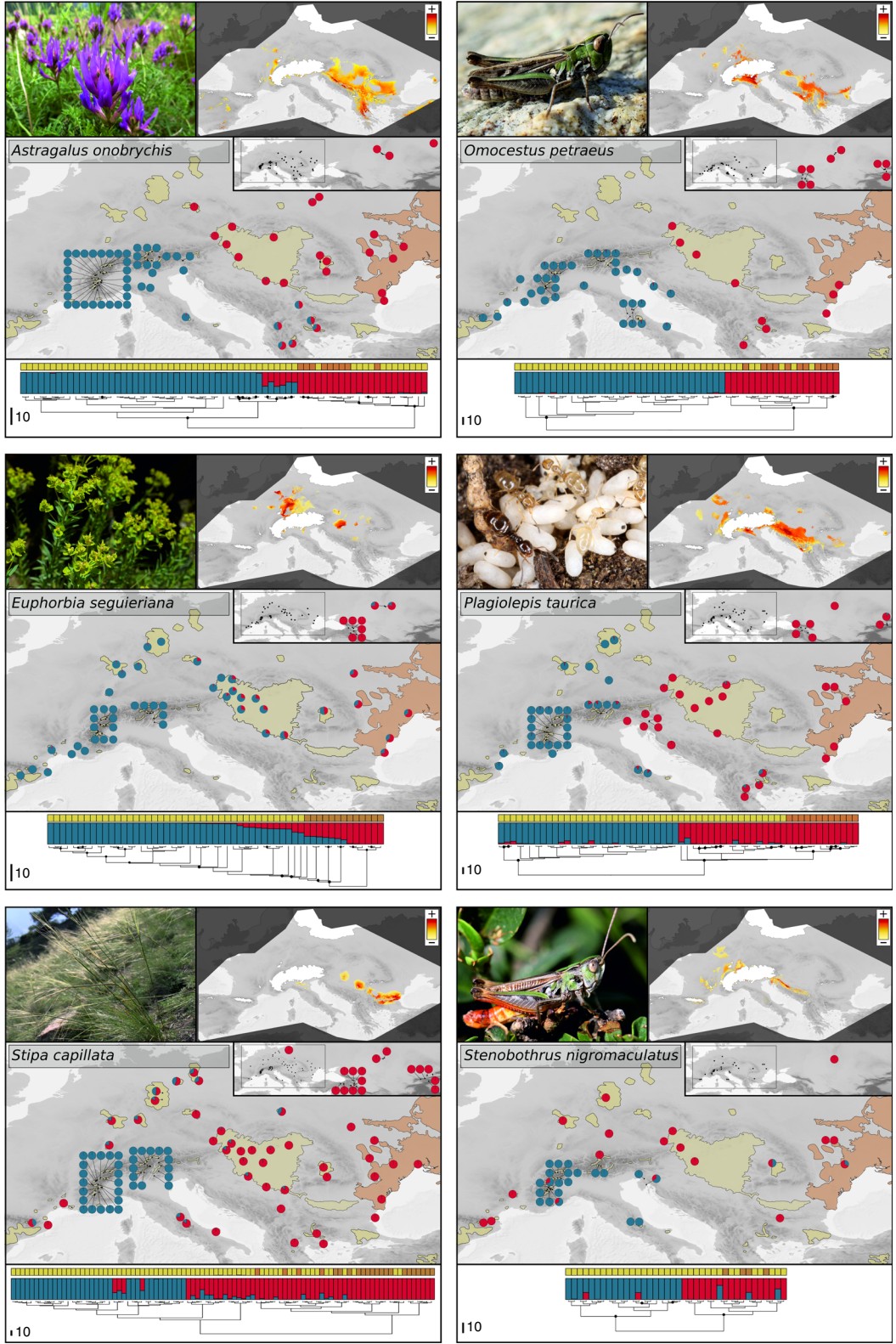

**Fig. 3 Distribution of lineages, genetic clusters, and modelled refugia.** Patterns of genome-wide Restriction Associated DNA based divergence in six Eurasian steppe species illustrated as maximum likelihood phylogenies (nodes with bootstrap support >75% are indicated with a black dot) and STRUCTURE[45] bar plots (colours illustrate STRUCTURE-based gene pools). The maps show the STRUCTURE-based gene pools' geographic distributions. Coloured grids above the bar plots indicate the location of the sampled populations in the extrazonal (yellow) or zonal steppes (brown); the extents of extrazonal and zonal steppes are outlined via the layers used in Fig. 1a (yellow: extrazonal steppes, brown: zonal steppes; not provided in the small inset showing exclusively zonal localities). Maps in the upper-right corner of each panel show potential refugia for cold and warm stages resulting from ecological niche modelling. Projected logistic probabilities of suitability above the species-specific Maximum Training Sensitivity Plus Specificity thresholds are indicated in colour. Source data are provided as a Source Data file.

Bayesian clustering pattern did not fully correspond to the phylogenetic main lineages. Here, one genetic cluster consisted of exclusively Alpine populations, while the second cluster comprised all other populations, including all zonal and non-Alpine extrazonal ones. The latter populations and the Alpine populations also showed local admixture, suggesting complex postglacial range dynamics. In *E. seguieriana*, a wide zone of admixed zonal and easternmost extrazonal populations connected the other extrazonal and the phylogenetically divergent Caucasian populations (Fig. 3).

**Mitochondrial DNA sequences reveal splits in the early Pleistocene.** The mitochondrial Cytochrome Oxidase 1 (COI) gene was sequenced from the grasshopper *O. petraeus* and the ant *P. taurica* (Supplementary Table 2). The topologies of COI-based *P. taurica* and *O. petraeus* phylogenies were largely congruent with RADseq-based phylogenies, that is, the main splits into a zonal and an extrazonal lineage were resolved similarly and were well-supported (Supplementary Fig. 3). Molecular dating placed the splits between zonal and extrazonal lineages in the mid to early Pleistocene, at 1.5 million years ago (Mya) (95% highest posterior density (HPD) 0.92–2.08 Mya) in *P. taurica* and at 2.46 Mya (95% HPD 1.0–4.52 Mya) in *O. petraeus* (Supplementary Fig. 3).

**Ecological niche modelling of current and past distributions.** Predictive performance of the niche modelling runs, as indicated by the area-under-the-curve, was good for all species; none of the models was strongly affected by overfitting as indicated by omission rates approaching zero (Supplementary Table 3). The predicted extant distributions of the six species fitted well to their actual distributions. In addition, some areas currently not harbouring any of the individual species received high suitability values (e.g. southern Denmark for *O. petraeus*, *S. nigromaculatus*, and *E. seguieriana*, Supplementary Fig. 4). Intriguingly, the zonal steppe habitats (e.g. western Ukraine) were assigned relatively low suitability values (Supplementary Fig. 4). Model response curves indicated that temperature variables [e.g. *Mean Diurnal Range* (bio2), *Temperature Seasonality* (bio4)], and occasionally *Precipitation Seasonality* (bio15), surpassed significant environmental suitability gradients at these zonal locations (not shown). In other words, those zonal localities were predicted as unsuitable because their climatic conditions differed considerably from the extrazonal conditions. This niche difference was confirmed using the background test developed by McCormack et al.[19]. The background tests revealed significant niche divergence between zonal and extrazonal localities of the six species (Supplementary Table 4, Supplementary Methods, Supplementary Note).

The MESS (multivariate environmental similarity surfaces)[20] analyses (Supplementary Fig. 5) showed a moderate (Community Climate System Model, CCSM3)[21] to high (Model for Interdisciplinary Research On Climate, MIROC)[214] similarity, that is, transferability, between the variables of the species localities under the presently observed climate that were used to train the MAXENT model, and the variables under the climate of the Last Glacial Maximum. Species-specific ENM-derived areas of stability indicated suitable habitats in extrazonal Europe throughout the Quaternary glacial cycles (Fig. 3). Areas of stability were found along the margin of the Alps outside the maximum extent of the Alpine glaciers during the Last Glacial Maximum for all species. These areas of stability were never connected to the larger ones modelled for the Balkan Peninsula (e.g. Dinaric Alps: *A. onobrychis*, *E. seguieriana*, *O. petraeus*, *P. taurica*, *S. nigromaculatus*) and the Pannonian basin (*A. onobrychis*, *E. seguieriana*, *S. capillata*). Except for *S. capillata*, small and isolated areas of

stability were projected not only for the southern but also for the northern and western margin of the Alps.

**Larger PD in the extrazonal steppes.** PD is an effective measure to assess conservation value by revealing unknown diversity patterns and unanticipated evolutionary processes irrespective of any taxonomic classification;[22] it is therefore a powerful and widely used measure to inform conservation strategies[23]. As such, we interpret PD as a benchmark for diversity and evolutionary processes in the framework of the Eurasian steppes. The extrazonal steppes harboured substantially higher PD than the zonal steppes, as shown by comparative rarefaction of PD (Fig. 4). Also, PD complementarity (i.e. the amount of unique PD represented by a subset of a phylogenetic tree that is not present in a reference set) was found to be larger in the extrazonal steppes compared with the zonal steppes in all six taxa (Supplementary table 5).

Within the extrazonal steppes and compared across all taxa, PD complementarity was evenly distributed, and maximum values were not aggregated in or restricted to single areas (Fig. 4). A general cross-species pattern emerged, in which the Alps, southern France, and the Apennine Peninsula harboured elevated PD complementarity (Fig. 4). The same was observed for the steppe relicts north of the Alps and on the northwesternmost Balkan Peninsula. However, the location of PD complementarity maxima within the mentioned regions was species specific (*E. seguieriana*: Western Alps, *S. capillata*: Western Alps, *O. petraeus*: southern France, *P. taurica*: central Germany, *S. nigromaculatus*: Apennine Peninsula; Fig. 4). Only one species, *A. onobrychis*, deviated from this overall pattern and had its largest PD complementarity on the southern Balkan Peninsula. While PD complementarity was relatively evenly distributed over the extrazonal steppes in *S. capillata*, *P. taurica*, and *E. seguieriana*, it was aggregated in the areas mentioned above in *A. onobrychis*, *O. petraeus*, and *S. nigromaculatus*.

## Discussion

The biogeography of the Eurasian steppes, and in particular the origin of the extrazonal European steppe biota, has fascinated biologists since the end of the 19th century[12]. Here, we present a comparative study on the phylogeography of the Eurasian steppes using a multidisciplinary, range-wide, multi-taxa, and cross-phyla approach that includes populations of six representative animal and plant steppe species sampled from Western Europe to Central Asia. Our results significantly contribute to understanding the spatio-temporal evolution of the Eurasian steppes and emphasise the importance of the European extrazonal steppes for the conservation and management of the steppe biome.

We found that the evolutionary history of the extrazonal European steppe biota has been largely independent from their zonal relatives, likely since the very onset of the Pleistocene glaciation cycles. Most extrazonal populations belong to lineages entirely absent from the zonal steppes, and admixture between extrazonal and zonal lineages is rare and localised (Fig. 2a, b, Fig. 3). These findings contrast a recent colonisation scenario, under which extrazonal populations of steppe species are young descendants of zonal steppe lineages that expanded into Europe during cold stages. Instead, we suggest a scenario of independent evolution of the extrazonal steppe biota in vicariance. With the exception of the plant *E. seguieriana*, such a vicariance scenario was observed across all studied species and might apply to many other Eurasian steppe species.

In two species, the ant *P. taurica* and the grasshopper *O. petraeus*, the deepest splits were roughly dated to the onset of the Pleistocene glaciations (Fig. 2c, Supplementary Fig. 3), which is in line with Pleistocene diversification within *E. seguieriana*[18] and

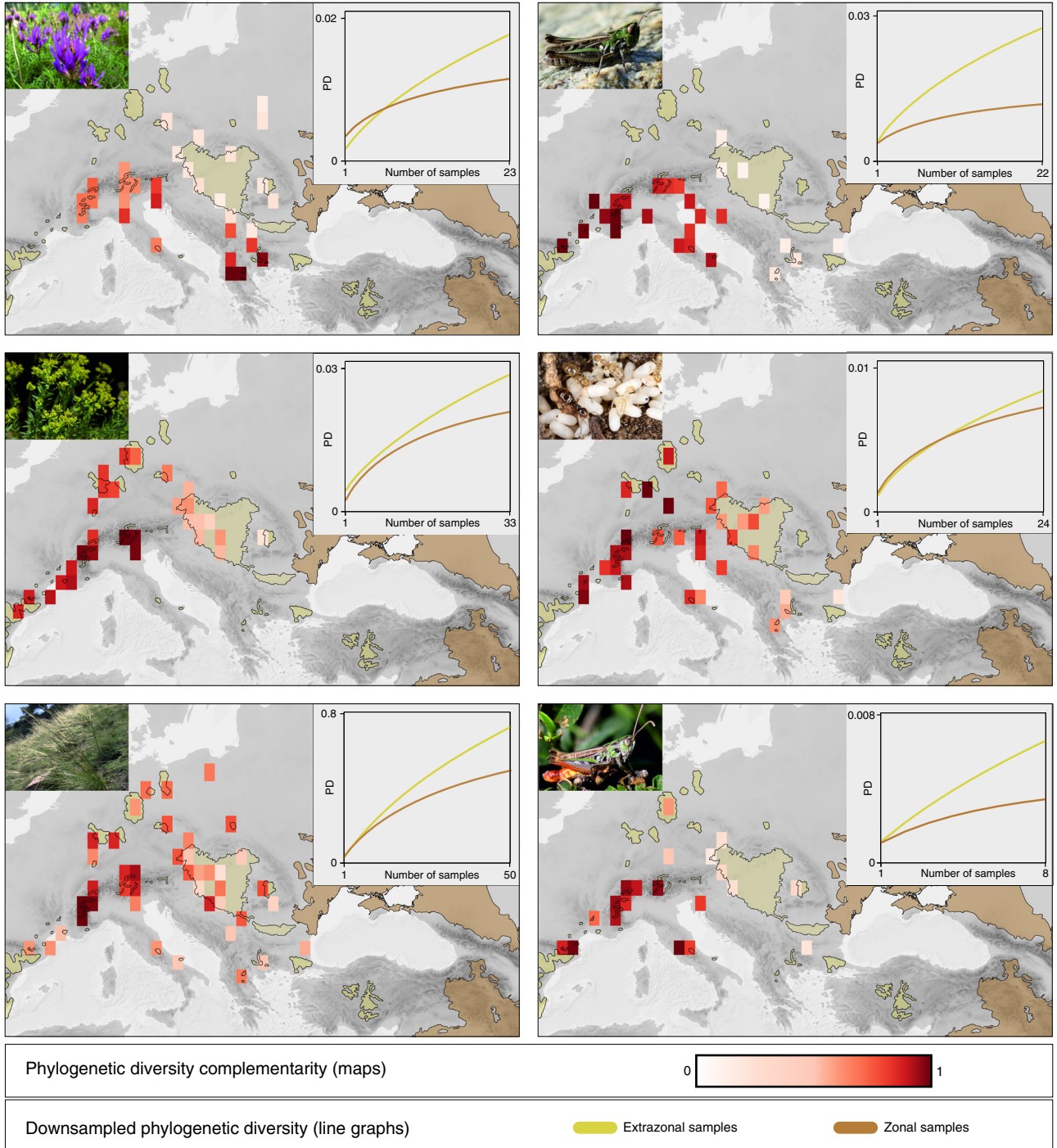

**Fig. 4 Distribution of phylogenetic diversity (PD) complementarity and PD in six Eurasian steppe species.** Grid cells of an equal edge length of one degree encompass sampled populations. PD complementarity values are shown as colour of heat. Curves in the insets show the rarefied PD of extrazonal and zonal populations; the extent of extrazonal and zonal steppes are outlined using the same layers as in Fig. 1a (yellow line: extrazonal steppes, brown line: zonal steppes). Source data are provided as a Source Data file.

emphasises the role of glacial cycles in shaping the genetic legacy of Europe's extrazonal steppes. In contrast to Europe's temperate forest taxa, for which the onset of the Pleistocene glaciations initiated a period of elevated extinction rate[24,25] and led to a decrease of genetic diversity north of the Alps[25,26], cold-tolerant steppe species were more widely distributed and abundant during Pleistocene cold stages and retreated during warm interglacials and in the Holocene[27,28]. In other words, Pleistocene climate oscillations and the interplay of range-contractions and range

expansions ultimately fostered an increase of genetic diversity and differentiation in the extrazonal steppe biota.

Pleistocene refugia harbour increased genetic diversity and rare alleles and should therefore be of particular interest for conservation[29,30]. Our results imply the existence of stable warm- and cold-stage refugia for extrazonal steppe biota apart from areas in today's zonal steppes (Fig. 2d). By modelling areas of stability, a powerful approach to predict long-term stable refugia[18,31], we identified multiple refugia throughout Europe

(Fig. 3). Based on our data, range expansions of extrazonal lineages occurred exclusively from isolated, sometimes very small refugia (e.g. *S. capillata*; Fig. 3) and not from the more continuous steppes in the Pannonian basin or the Pontic plains as suggested previously (Fig. 3)[11,32]. Thus, we highlight the role of extra-Pannonian refugia for the evolution and diversification of European steppe biota and emphasise their importance for conservation.

The extrazonal steppes, albeit much smaller than the zonal steppes, were found to harbour elevated PD, a larger amount of unique PD, and endemic lineages (Fig. 2e, f, Supplementary Table 5). This is because the steppe outposts west of the zonal steppe belt did not only act as the cradle and sole genetic reservoir for exclusively extrazonal steppe lineages but also harboured disjoint populations of zonal lineages (Fig. 3). Thus, the extrazonal steppes also harbour a larger intraspecific PD than the zonal steppes (Fig. 4). Following the logics of modern-day conservation policy, in which societal consent and continuous funding are central factors to ensure long-term success of any conservation strategy, we focused on identifying areas of elevated PD complementarity in the extrazonal steppes that could be prioritised in conservation practice[22,33,34] (Fig. 2f). We show that PD complementarity is generally larger in the extrazonal steppes compared with the zonal steppes and, within the extrazonal steppes, not aggregated in or restricted to single areas; it is, in fact, rather evenly distributed across the extrazonal steppes and exhibits largely species-specific spatial patterns (Fig. 4, Supplementary Table 5). This again emphasises the uniqueness and conservation value of extrazonal steppes compared with the zonal steppes as no single region exhibited pervasively elevated PD (Fig. 2f, Fig. 4). Given this and the restriction of extrazonal lineages to the steppe outposts, we emphasise that conservation efforts should not concentrate on selected individual areas but rather on the European extrazonal steppes as a whole.

In all, the extrazonal European steppes fulfil all previously defined criteria (Fig. 2) for areas of outstanding conservation value, that is, occurrence of private, geographically restricted phylogenetic lineages and genetic clusters (Fig. 2a, b), deep phylogenetic splits that suggest long-term independence and ancient age (Fig. 2c), occurrence of multiple stable refugia (Fig. 2d), high and at the same time unique PD (Fig. 2e, Supplementary Table 5), and species-specific, idiosyncratic distribution of this PD (Fig. 2f). To conserve the genetic legacy of the Eurasian steppe biome as a whole, each steppe outpost has to be treated as a significant and largely independent entity that evolved in isolation for most of its history. Moreover, we argue that the significance of our results extends beyond the conservation of the Eurasian steppe biome. They emphasise the need of cross-phyla, comparative studies that focus on multiple taxonomic groups when evaluating the actual conservation value of a biome. "Classical" biogeographic hypotheses, such as the presumably young age of European extrazonal steppes, need to be re-evaluated on a case-by-case basis, and conservation decisions must be taken by considering multiple taxa across different taxonomic, evolutionary, and ecological levels.

## Methods

**Sampling**. Six species were selected, which are characteristic, widespread, abundant, and often co-occurring elements of the Eurasian steppes. In addition, heteroploid, taxonomically difficult species were avoided, the exception being *A. onobrychis*. Samples of the plant species *A. onobrychis* L. (Fabaceae), *E. seguieriana* Neck. (Euphorbiaceae), and *S. capillata* L. (Poaceae), and the animal species *P. taurica* Santschi, 1920 (Formicidae), *O. petraeus* (Brisout de Barneville, 1856), and *S. nigromaculatus* (Herrich-Schäffer, 1840) (both Acrididae) were collected between 2014 and 2017. All species were sampled across their distribution ranges, while extrazonal occurrences were more densely sampled than zonal ones. Of the six study species, only *S. capillata* was suggested to be widespread in the Iberian Peninsula. There, *S. capillata* was not exhaustively sampled outside the Pyrenees, as preliminary data suggested that the Iberian populations belong to another, cryptic

species. This divergence is supported by the classification of the Iberian steppes as Mediterranean, instead of Central Asian, vegetation type[35]. In total, 456 populations from 320 localities were sampled (details are given in Supplementary Data 1). Identification of animals at the species level was done using the corresponding keys (grasshoppers[36], ants[37]). Collected specimens were stored in silica gel (plants) or 96% ethanol (animals) for further analyses; herbarium vouchers and animal specimens are stored at the Departments of Botany (herbarium IB) and of Ecology, University of Innsbruck, respectively.

**DNA extraction and RADseq library preparation**. Plant DNA was extracted from leaf tissue with a sorbitol/high-salt cetyltrimethylammonium bromide method[38] and purified using the NucleoSpin gDNA clean-up kit (Macherey-Nagel, Düren, Germany). Animal DNA was extracted from leg muscle tissue (*O. petraeus* and *S. nigromaculatus*) or whole animals (*P. taurica*) with the DNeasy Blood & Tissue Kit (Qiagen, Düsseldorf, Germany). Altogether, 370 populations from 266 localities were sequenced by RADseq (for details see Supplementary Table 1, Supplementary Table 6). Prior to the RADseq analyses, the genome size of each species was determined with flow cytometry to aid the selection of suitable restriction enzymes[39]. RADseq libraries were prepared using published protocols with minor modifications[40] (for details see Supplementary Material).

**Identification of RADseq loci and SNP calling**. The raw reads were quality filtered and demultiplexed according to individual barcodes using Picard BamIndexDecoder (included in the Picard Illumina2bam package; available from https://github.com/wtsi-npg/illumina2bam) and the program process_radtags.pl implemented in Stacks[41]. All resulting raw RADseq data are available in the NCBI Short Read Archive (accession numbers in Supplementary Data 1).

Before calling SNPs from the demultiplexed Illumina reads, the data were species-specifically preprocessed. Large genomes, as observed in *O. petraeus* and *S. nigromaculatus*, are known to contain large portions of pseudogenes, transposable elements, and noncoding DNA[42]. These elements are prone to cause problems in SNP calling procedures, as homology of a fragment cannot be ensured if multiple copies are present in an organism's genome. To address this issue, reads of *O. petraeus* and *S. nigromaculatus* were mapped to a reference genome. To do so, RepeatMasker (A.F.A. Smit, R. Hubley & P. Green RepeatMasker at http://repeatmasker.org) was used to identify and mask repeated elements present in the *Locusta migratoria* genome[42] (GenBank: AVCP000000000.1). In the next step, the quality-filtered RADseq reads were mapped to the masked *L. migratoria* genome using Stampy v. 1.0.20[43]. Mapped reads were selected with Samtools[44], and only those were further used in downstream analyses, starting with ref_map.pl of Stacks[41].

Apart from the two grasshopper taxa, RAD loci were assembled, and single nucleotide polymorphisms (SNPs) were called de novo, using the denovo_map.pl pipeline in Stacks version 1.46[41]. Settings for the denovo_map.pl program [i.e. a maximum number of differences between two stacks in a locus in each sample (−M), and a maximum number of differences among loci to be considered as orthologous across multiple samples (−n)] were separately evaluated and optimised for each species, considering the number of potentially paralogous loci based on trial analyses (Supplementary Table 1).

The function export_sql.pl in the Stacks package[41] was used to extract loci information from the catalogue. RAD tags were removed if (i) in any of the samples more than two alleles were detected, to reduce the risk of including paralogues in the dataset; (ii) the number of deleveraged tags was higher than 0. The programme populations implemented in the software Stacks[41] was used to export the selected loci, whereas whitelists were used to exclude the unwanted loci as described above. For the first dataset, later used for Bayesian clustering, a set of RAD loci was exported into STRUCTURE[45] format using the --structure and --write_random_snp flags[41]. The latter command was used to select only a single, random SNP per fragment to minimise the chance of selecting linked loci for the Bayesian clustering. For this dataset, all individuals sampled from the same site were defined as a population. For a second dataset per species, later used for phylogenetic tree reconstruction, concatenated RAD loci were exported as sequence alignment in phylip format using the --phylip_var_all flag in Stacks[41].

**Genetic clustering and phylogenetic reconstruction based on RADseq data**. The optimal grouping of the populations was calculated using Bayesian clustering in STRUCTURE (version 2.3.4), using the admixture model with uncorrelated allele frequencies[45]. The analyses were performed separately for each species with input data characteristics specified in Supplementary Table 1 and the number of analysed individuals and populations shown in Supplementary Table 6. Ten replicate runs for K (number of groups) ranging from 1 to 10 were carried out using a burn-in of 200,000 iterations followed by 2,000,000 additional MCMC iterations. The optimal number of groups (K) was identified by identifying where the increase in likelihood started to flatten out, the results of replicate runs were similar, and the clusters were non-empty. Additionally, the deltaK criterion was employed, reflecting an abrupt change in likelihood of runs at different K[46].

The concatenated RAD-locus alignments of each species were analysed as follows: Phylip alignments were imported into R (version 3.4.4) using the R package phrynomics (https://github.com/bbanbury/phrynomics.git). Subsequently,

loci and individuals that contained more than 75% missing or ambiguous base calls at polymorphic sites (more than 85% in *O. petraeus*) were removed for downstream phylogenetic analyses (Supplementary Table 1). This step was done to meet the criteria proposed for robust phylogenetic inference from SNP data[47]. To infer phylogenetic relationships, maximum likelihood (ML) phylogenies were calculated using RAxML (version 8.2.8)[48]. All tree searches were done under a general time reversible model (GTR) with categorical optimisation of substitution rates (ASC_GTRCAT)[48]. Optimal substitution models were selected beforehand via the smart model-selection algorithm[49]. The best-scoring ML trees were bootstrapped using the frequency-based stopping criterion[50]. All phylogenies were initially rooted using an outgroup (Supplementary Table 7, not shown in the final figures). As the final alignments used for phylogenetic inference contained only variant sites, Felsenstein's ascertainment bias correction implemented in RaxML was used to account for the missing invariant sites as recommended[51]. In addition to these complete phylogenies, in which all accessions passing the defined criteria were used, an additional population phylogeny, containing only one random individual per sampled population, was calculated for each species, following the same methodology. These population trees were later used to display the relationships between sampled populations (Fig. 3). The number of populations and individuals used to infer both phylogenies is given in Supplementary Data 1 and Supplementary Table 6.

**Ecological niche modelling and paleo-projections**. To avoid background selection and sampling bias, sparsely sampled Eurasia east of the Crimea was excluded for ENM[52–54]. Model overfitting due to spatial autocorrelation[55] was reduced by removing localities within a spherical distance of 5 km using PASSaGE2 v. 2.0.11.6[56]. The final dataset comprised 380 records (details in Supplementary Data 1, Supplementary Table 6). Bioclim variables for current climatic conditions were obtained from Worldclim v.1.4 (http://www.worldclim.org/)[57] at a resolution of 30 arc-seconds and clipped to the area of study encompassing the European distribution of the six steppe species. To compensate Worldclim's low precision for precipitation variables in mountainous areas[57], observational data of 23 alpine climate stations were used to generate interpolated precipitation layers for the Alps (see Supplementary Methods, Supplementary Fig. 6, Supplementary Table 8). Pairwise correlation between variables was assessed using ENMTools (version 1.4.4)[58], and variables with a Pearson's correlation coefficient > 0.9 were removed based on expert knowledge. Two variables (bio18, bio19) were excluded for technical reasons (Supplementary Methods). For the final models, eleven variables were used (bio2, bio3, bio4, bio8, bio9, bio10, bio11, bio12, bio15, bio16, bio17; see Supplementary Methods). All ENMs were generated using MAXENT (version 3.3.3.k)[59]. Model parameters and details on the replication method are described in Supplementary Methods.

Species-specific tuning and jackknife tests (for details, see Supplementary Methods, Supplementary Table 3) were done to improve model performance and transferability[60]. The resulting models were projected to conditions of the Last Glacial Maximum, based on MIROC3.2[21] and CCSM3[21], at 30 arc-seconds spatial resolution. The projected variables were restricted to values encountered in model calibration under current conditions using clamping. To quantitatively assess projection uncertainty, MESS were used, whereby negative and positive MESS scores indicate non-analogue and analogue climates, respectively[20]. To reduce uncertainty stemming from different models, consensus prediction maps were generated by averaging over both paleo-projections (CCSM and MIROC)[61]. Areas of Stability were defined as grid cells that received higher suitability than the Maximum Training Sensitivity Plus Specificity (MTSS) threshold and were not covered by ice during Pleistocene cold stages[62]. Average Suitability of Stable Area maps were calculated as mean suitability of the individual consensus predictions after the MTSS had been applied. Projections and MESS analyses of all study species are presented in Supplementary Fig. 5.

**PD and spatial patterns of diversity**. To evaluate how much of each species' diversity was represented by its extrazonal distribution compared with its zonal distribution, PD was calculated for the respective area[17]. To account for uneven sampling between the zonal and extrazonal steppes, the rarefaction approach implemented in the R package phylorare was used[63]. For these calculations, the complete maximum likelihood phylogenies were used (see section above).

PD complementarity was put in a spatial context to assess the uniqueness and heterogeneity of the extrazonal steppes. To do so, geographic entities, that is, grid cells of one-degree edge length (i.e. approximately 100 km), were defined for the extrazonal steppes prior to the analysis. For each of these grid cells, PD complementarity was calculated (i.e. the branch length represented within the grid compared with a reference set of branches)[17]. This was done based on the complete ML phylogenies (see section above), using all branches occurring in the zonal steppes as a reference. Because of the sampling-related unequal number of branches per grid cell, a random downsampling approach was implemented. In this stepwise procedure, the full dataset of each taxon was randomly reduced in such a way that the downsampled dataset still contained all grid cells but only one single branch per grid cell. For each of these reduced data sets, PD complementarity was calculated using all zonal branches as reference. After repeating this step 100 times per taxon, median PD complementarity values were calculated for each grid cell. To enable comparability across taxa, these values were transformed via division by

the respective maximum median value obtained for a particular taxon so that the final values ranged between 0 and 1. Downsampling was done via custom R scripts, and the calculation of all PD complementarity was done in PDA (version 1.0.3)[64]. Additionally, PD complementarity for both all extrazonal steppes and all zonal steppes was calculated for each taxon. This was done to compare the amount of unique PD within each area, again using PDA (version 1.0.3)[64].

**Mitochondrial DNA sequencing**. For two species, *O. petraeus* and *P. taurica*, parts of the mitochondrial COI gene were sequenced. Mitochondrial DNA sequencing in grasshoppers is prone to pseudogene amplification[65]. Hence, species-specific primers were designed using a cDNA template of the COI gene. To do so, total RNA was extracted from *O. petraeus* hind femur tissue, which had been stored at −80 °C immediately after field collection, using the Qiagen RNeasy Micro Kit (Qiagen, Düsseldorf, Germany). Extracted RNA was transcribed into cDNA via RevertAid reverse transcriptase (Thermo Fisher Scientific) in a 60-minute incubation step at 42 °C followed by an enzyme-deactivation step at 70 °C for 10 minutes. Using the cDNA as a template, the COI gene was amplified using published primers under standard PCR conditions (Supplementary Table 2). The resulting fragment was used to design specific primers inwards to the original binding sites. Primer quality and specificity were evaluated in silico using fastpcr (version 6.0)[66]. Specific primers were then used to amplify the COI gene from the DNA extracts previously prepared for RADseq using the Rotor-Gene Probe PCR Master mix (Qiagen, Düsseldorf, Germany) (Supplementary Table 2). For amplification of COI of *P. taurica*, published standard primers were used (Supplementary Table 2). Oligo-nucleotide synthesising and all Sanger sequencing steps were done by Eurofins Genomics, Ebersberg, Germany. All generated sequences were uploaded to NCBI GenBank (accession numbers in Supplementary Data 1).

**Molecular dating of divergence based on mitochondrial DNA sequences**. A mitochondrial mutation rate of 0.01615 substitutions per site per million years (My), which equals 3.2% divergence per My, previously proposed for European arthropods[67], was used to date divergence events in the COI-based phylogenies of *P. taurica* and *O. petraeus*. A Bayesian uncorrelated lognormal relaxed clock as implemented in BEAST v.1.8[68] was applied to estimate the branch lengths. Before, the best substitution model, TrN+G for *P. taurica* and GTR + I + G for *O. petraeus*, was identified with jModelTest v.2.1.4[69] by using the corrected Akaike Information Criterion. The clock rate was modelled with a lognormal distribution and a mean of 0.016 substitutions per site per My, a standard deviation of 0.2 and an offset at 0.0[67]. Two independent analyses were run for a total of 10,000,000 generations. Log files were analysed using TRACER v.1.5[70] to assess convergence and to ensure that the effective sample size (ESS) for all parameters was >200. The resulting trees were combined using LogCombiner v.1.7.5[68] with a burn-in of 25%. Subsequently, a maximum clade credibility tree was constructed using TreeAn-notator v.1.7.5[70].

**Reporting summary**. Further information on research design is available in the Nature Research Reporting Summary linked to this article.

## Data availability
Demultiplexed RADseq sequencing data and mitochondrial DNA sequences are available from the NCBI GenBank Short Read Archive and NCBI Nucleotide Database, respectively (accession numbers in Supplementary Data 1, Supplementary Table 7). Source data underlying Figs. 1, 3, and 4 and Supplementary Fig. 4 are provided as Source Data file.

## Code availability
The code that was used to randomly subsample tips from a phylogenetic tree and calculate phylogenetic diversity complementarity values has been made publicly available on GitHub: https://github.com/philippkirschner/PD_downsampler.

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

## Acknowledgements

The authors especially thank all collectors listed in Supplementary Data 1 and all colleagues that provided locality data. P. Andesner, J. Baar, A. Kluibenschedl, M. Magauer, D. Pirkebner, and E. Zangerl helped with laboratory work, and M. Gassner helped with preparation of some figures. We thank C. Lebas for pictures of *P. taurica*. We are thankful to S. Laffan and D. Rosauer for constructive discussion. Collecting permits were issued by the Autonomous Province Bozen, Italy (No. 333338), the Autonomous Region Vallé d'Aosta, Italy (14191/Rr), the Austrian federal state Burgenland (5-N-A1007/586-2014), the Austrian federal state Lower Austria (RU5-BE-1049/001-2014), the German federal state Rheinland-Pfalz (Nord: 425-104.141.1402, South: 42/553-251), the German federal state Thuringia, and the canton Grisons, Switzerland (AV-2014-210). The present study was funded by the Austrian Science Fund (FWF, project P25955 "Origin of steppe flora and fauna in inner-Alpine dry valleys" to P.S.) and the Tiroler Wissenschaftsfonds (TWF, UNI-0404/2066, "Comparing information efficiency of high- versus low-resolution genome scans for phylogeographic studies" to P.K. and TWF, UNI-0404/1642 "Immigration history of the steppe species *Euphorbia seguieriana* in inner-Alpine dry valleys and its phylogenetic position within *Euphorbia* sect. *Pithyusa*" to B.F.). The computational results presented have been achieved in part using the HPC infrastructure LEO of the University of Innsbruck and in part using the Vienna Scientific Cluster (VSC).

## Author contributions

P.S., F.M.S. and A.H. conceived the project and designed the study. P.K., E.Z, F.M.S. and P.S. co-wrote the manuscript. O.P., E.T. and E.Z. adapted and fine-tuned the RADseq protocol. P.K. and E.Z. performed library preparation, RADseq mapping and most analyses. A.G. performed niche modelling analyses and wrote corresponding parts in the manuscript. A.H., B.F., P.K., B.C.S-S., P.S., F.M.S., E.T., and E.Z. collected samples in the field. W.A., B.F., A.G., A.H., O.P., I.S., B.C.S-S., and E.T. contributed to the development of the manuscript and improved earlier drafts of the paper.

## Competing interests

The authors declare no competing interests.

## Additional information

## The STEPPE Consortium

Wolfgang Arthofer[1], Božo Frajman[2], Alexander Gamisch[3], Andreas Hilpold[4], Philipp Kirschner[1], Ovidiu Paun[6], Isabel Sanmartín[7], Birgit C. Schlick-Steiner[1], Peter Schönswetter[2], Florian M. Steiner[1], Emiliano Trucchi[5] & Eliška Záveská[2]

