## [Peer Review File · Nature Communications]

Reviewers' comments:

Reviewer #1 (Remarks to the Author):

The topic and context are very important - the authors note "The Eurasian steppes are the second-largest continuous biome on Earth" and that "Eurasian steppes are also one of the most threatened biomes worldwide".

The rationale for the study is "extrazonal steppe biota instead could have persisted past warm interglacials in refugia ... evolutionary processes such as genetic drift and/or ecological adaptation would have caused deep divergence among zonal and extrazonal lineages.....in the latter case, the conservation value of extrazonal steppe biota would increase considerably and render them far more important units of biodiversity conservation than previously assumed. .. resolving these two contrasting hypotheses could substantially change the current perspective on the conservation of extrazonal steppes."

The authors indeed find this, by appropriately looking at PD methods.

However, the actual analysis and its rationale needs to be tightened up – this would be helped by greater attention to precursor PD analyses. If this analysis aspect can be improved then the study could be a valuable exemplar for other studies in this important emerging area of biodiversity conservation.

1) PD and PD complementarity

Re " in five of the six investigated species, the extrazonal steppes harbour a larger intraspecific phylogenetic diversity than the zonal steppes"

This is a good key finding and is documented well in the figs (accum curves), but the case for conservation calls for more information – what PD does the extrazonal have that the zonal does not have? In the Faith 1992 PD study on the phylogeography level, these contributions are PD complementarity values and they are important to the conservation case.

So, regarding the key claims – "To conserve the genetic legacy of the Eurasian steppe biome as a whole, each steppe outpost has to be treated as a significant and largely independent entity that evolved in isolation for most of its history" and "conserving what is left of Europe's steppes is crucial for conserving the biological diversity of the entire Eurasian steppe biome" and "emphasize the importance of the European extrazonal steppes for the conservation and management of the steppe biome.":

Calculating PD complementarity values, for the whole extrazonal and for individual bits, would be informative about these claims.

And similarly, re

“To conserve the genetic legacy of the Eurasian steppe biome as a whole, each steppe outpost has to be treated as a significant and largely independent entity that evolved in isolation for most of its history.” We ask -

Where is the evidence for this unique evolution - how many distinct lineages etc are in these bits?

Related to this the claim “To evaluate how much of each species’ diversity was represented by its extrazonal distribution compared with its zonal distribution, phylogenetic diversity was calculated for the respective area⁶².” -

The two totals do not tell us about amount of PD overlap, nor unique PD within each.

Such analysis might strengthen the claim that “We found that the evolutionary history of the extrazonal European steppe biota has been largely independent from their zonal relatives, likely since the very onset of the Pleistocene glaciation cycles. Most extrazonal populations belong to lineages entirely absent from the zonal steppes.....”

Are there lineages restricted or nearly-restricted to each? It is not revealed in the current analysis. (and Fig 3 is referred to (with unclear use of blue vs red) but seems to cover only extrazonal – how does this inform at all about contrast with zonal?)

Example PD analysis (phylogeography for Europe) can be found in Faith 1992 and later papers.

2) endemism and evol. potential

The CANAPE related analysis of PD endemism do not appear to be informative about conservation value, and do not have any clear link to conservation priority regarding evol potential.

The authors say “Following the logics of modern day conservation policy, in which societal consent and continuous funding are central factors to ensure long-term success of any conservation strategy, we focused on identifying European steppe areas with significantly increased evolutionary potential^{21,34} (Figure 2F-G).

But fig 2F-G does not have any clear link to evol potential criteria,

And re

“the conservation value of zonal versus extrazonal areas using the concept of phylogenetic endemism^{17,18}

“phylogenetic endemism exhibited species-specific spatial patterns (Figure 4). This suggests that, in order to conserve the evolutionary potential^{16,19,21} of the extrazonal steppe biota,....

Again, I do not think that these indices of phylogenetic endemism can be defended as indicating anything about evol potential.

This conclusion therefore seems unwarranted:

“Altogether, phylogenetic endemism exhibited species-specific spatial patterns (Figure 4). This suggests that, in order to conserve the evolutionary potential^{16,19,21} of the extrazonal steppe biota, conservation efforts should not concentrate on selected individual areas but rather on the European extrazonal steppes as a whole, as there is no single region of pervasively elevated conservation value.”

And this conclusion is misleading “The extrazonal steppes, albeit much smaller than the zonal steppes, were found to harbour both elevated phylogenetic diversity and endemic lineages (Figure 2E-F).”

But the analysis does not explore possible lineages actually restricted (endemic) to extrazonal – a revised analysis could do this.

3)

This diagram and legend is Unclear: “Figure 2. ...A, depth of phylogenetic split between observed clusters

For fig 3 - Sadly, it is not made clear what red vs blue dots are in this critical fig ...it says “(colours illustrate gene pools).” But “gene pool” is never mentioned in the text of the paper

My Recommendations:

Draw on the early PD phylogeography analyses that made use of PD complementarity and endemism (Faith 1992)

and also see Faith 2008 chapter in Conservation Biology: Evolution in Action edited by Scott P. Carroll, Charles W. Fox, for a guideline study, including exploring congruence among trees in geo patterns – extract attached

Also, the value of PD in conservation could be better highlighted for the reader – e.g. it is used by IPBES and the EDGE program. (see e.g. <https://danielpfaith.wordpress.com/pages>)

Dan Faith

Reviewer #2 (Remarks to the Author):

European steppe outposts harbour a high biodiversity and are highly threatened by current land-use change. However, for nature conservation one might argue that they are irrelevant because their area was always below 1% of that of the zonal steppes. Accordingly, deeper cross-taxa insights into the evolutionary history and genetic diversity of these outposts are highly needed.

All in all, the manuscript is novel, well written and the results are sound. It is an important contribution for the conservation of European steppe outposts in times of global change.

Minor remarks

Line 35. Add "." after scale

Reviewer #3 (Remarks to the Author):

This is an excellent study on the European steppe's biota which, to my knowledge, is providing novel and very significant results. I am happy to see that they have treated the issue through a multidisciplinary methodology (ecological and molecular markers) and, in addition to provide results that are of interest themselves from the biological point of view, they would have also direct implications in conservation policies. The authors are also using the most recent and cutting-edge methodologies, the analyses are in general technically sound and, to my opinion, manuscript has the potential to be an important piece of knowledge to ecologists and biodiversity managers. Therefore, my suggestion is that the ms. has the potential to be published in one of the world-leading multidisciplinary journals such as Nature Communications. I have, however, several comments and suggestions that I hope may help the editors to reach a final decision. Specifically, these are:

1. Introduction, lines 70-74. You have used six species (three plants and three animals) as a representative sample of the steppe biome. The number of species selected is a somewhat arbitrary criteria (to me, enough if they are actually representative of different life-history traits), but the reasons why these six species and not other ones are selected should be explained here. It is also a pity that the species chosen by you are not very representative of the most peripheral region that contain large patches of extrazonal steppe: the Iberian Peninsula. According to Fig. 1A, at present this peninsula has relatively large stretches of steppe that constitute the westernmost tip of this biome, and thus deserving of being sampled more extensively. However, I think that the study is still providing very valuable results, so this should not be regarded in any way as a "fatal" flaw.

2. The sampling of the material for genetic analyses is generally sound, with an adequate coverage of the geographical areas where the species occur (perhaps with the relative exception of *Stipa capillata*; see my comments below) as well as sampling sizes. However, I am very curious why the sample sizes for ENM are so low (even after the 5-km radius removing of occurrences) for plants and animals that are relatively common in the European steppes (especially for the plants). For example, if you have a look to GBIF and focus on Europe, you will see that *Stipa capillata* has almost 4000 occurrences (<https://www.gbif.org/species/4143519>), *Astragalus onobrychis* about 3500 (<https://www.gbif.org/species/5342618>), and *Euphorbia seguierana* over 8000 (<https://www.gbif.org/species/3066179>). A reason may be that the authors have only used their own records (Supplementary Excel table). For example, *Stipa capillata* seems to have many localities throughout the Iberian Peninsula but only two locations are used. Why the authors have not expanded the ENM occurrences using other sources like biodiversity databases (e.g. GBIF, iNaturalist), articles, books, etc.?

3. Although I am not specialist, it seems that the genetic analyses based on RADseq data are adequate, as well as those based on mtDNA.

4. Material & Methods, lines 303-304. The number of MCMC is always a controversial question when running STRUCTURE, but most recent works (probably as a consequence of the increase of calculation capacity of modern computers) use at least one million MCMC. An additional suggestion is improving the number of runs.

5. Given that you have included additional information regarding the methods as supplementary material, some methodological questions on ENM should be included here: (1) on the basis of what criteria the 11 variables have been selected from the 19 ones (expert one?); (2) what replication method has been used with Maxent (bootstrap, subsample, cross-validation) and how many replicates have been run; (3) what interpolation method has been used to transform the bioclimatic variables from 2.5 arc-min (the resolution at which these variables are deposited in WorldClim database) to 30 sec.

6. Material & Methods, lines 335-337. Your efforts to correct the precipitation variables for mountain areas are commendable. It is a pity that such correction cannot be done for other mountain areas where problems of inaccuracy are probably the same, such as the Pyrenees, the Apennines or the Carpathians.

7. Results, lines 137-140. Finding traces of niche divergence between zonal and extrazonal occurrences is a very interesting and significant result, as this is congruent with the genetic data (i.e., evolutionary history of extrazonal steppes independent from zonal ones) as one may expect (see e.g. Xu et al., 2015, *Ann Bot.* 116: 35–48), and adds more conservation value to the extrazonal steppes. I strongly suggest to explore for this seemingly niche differentiation, by expanding the niche comparative analyses to the other five taxa and, if possible, moving to the E-space. The methodologies developed by several authors in recent years are versatile and can be applied to cases with low number of occurrences (but see my comment no. 2), including that of McCormack et al. (2010, *Evolution* 64: 1231–1244) and that of Broennimann et al. (2012, *Global Ecol. Biogeogr.* 21: 481–497).

8. Results, lines 163-165. To me, finding neoendemism only in extrazonal steppes is not surprising, giving the strong mountainous nature of the Alps (facilitating genetic isolation).

Point-by-point response to the referees' comments

REVIEWER 1

Reviewer 1, Comment 1.

PD and PD complementarity □

"(...)in five of the six investigated species, the extrazonal steppes harbour a larger intraspecific phylogenetic diversity than the zonal steppes"

This is a good key finding and is documented well in the figs (accum curves), but the case for conservation calls for more information – what PD does the extrazonal have that the zonal does not have? In the Faith 1992 PD study on the phylogeography level, these contributions are PD complementarity values and they are important to the conservation case.

So, regarding the key claims – "To conserve the genetic legacy of the Eurasian steppe biome as a whole, each steppe outpost has to be treated as a significant and largely independent entity that evolved in isolation for most of its history" and "conserving what is left of Europe's steppes is crucial for conserving the biological diversity of the entire Eurasian steppe biome" and "emphasize the importance of the European extrazonal steppes for the conservation and management of the steppe biome." □ Calculating PD complementarity values, for the whole extrazonal and for individual bits, would be informative about these claims.#

>>> Response 1.1. Done. We have found this suggestion very useful and feel that implementing it has strongly improved our paper – thank you. Thus, to show the extent of unique phylogenetic diversity (PD) in the extrazonal steppes, we have implemented additional analyses utilizing the concept of PD complementarity as suggested. We have calculated PD complementarity compared with the zonal lineages for all sampled parts of the extrazonal steppe and have summarized the results in the revised version of Figure 4. We now show the widespread occurrence of unique PD in the extrazonal steppes in all species. To put the data into a spatial context and to evaluate the conservation significance of "individual bits", we have decided to calculate PD complementarity for pre-defined geographic units (1x1° grid cells); to account for the unequal number of branches across the grid cells, a downsampling approach has been utilized.

The approach is described in detail in the methods section, and the relevant passage reads:

“PD complementarity was calculated and put in a spatial context to assess the uniqueness and heterogeneity of the extrazonal steppes. To do so, geographic entities, that is, grid cells of one-degree edge length (i.e. approximately 100 km), were defined for the extrazonal steppes prior to the analysis. For each of these grid cells, PD complementarity was calculated (i.e. the branch length represented within the grid compared with a reference set of branches)¹⁷. This was done based on the complete ML phylogenies (see section above), using all branches occurring in the zonal steppes as a reference. Because of the sampling-related unequal number of branches per grid cell, a random downsampling approach was implemented. In this stepwise procedure, the full data set of each taxon was randomly reduced in such a way that the downsampled data set still contained all grid cells but only one single branch per grid cell. For each of these reduced data sets, PD complementarity was calculated using all zonal branches as reference. After repeating this step 100 times per taxon, median PD complementarity values were calculated for each grid cell. To enable comparability across taxa, these values were transformed via division by the respective maximum median value obtained for a particular taxon so that the final values ranged between 0 and 1. Downsampling was done via custom R scripts, and the calculation of PD complementarity was done in PDA (version 1.0.3)⁶³.”

The corresponding passage in the Results section now reads:

“Within the extrazonal steppes and compared across all taxa, PD complementarity (i.e., unique PD as compared with the zonal steppes) was evenly distributed, and maximum values were not aggregated or restricted to single areas (Figure 4). A general cross-species pattern emerged, in which the Alps, southern France, and the Apennine Peninsula harbored elevated PD complementarity (Figure 4). The same was observed for the steppe relics north of the Alps and on the northwesternmost Balkan Peninsula. However, the location of PD complementarity maxima within the mentioned regions was species-specific (*E. seguieriana*: Western Alps, *S. capillata*: Western Alps, *O. petraeus*: southern France, *P. taurica*: central Germany, *S. nigromaculatus*: Apennine Peninsula; Figure 4). Only one species, *A. onobrychis*, deviated from this overall pattern and had its largest PD complementarity on the southern Balkan Peninsula. While PD complementarity was relatively evenly distributed over the extrazonal steppes in *S. capillata*, *P. taurica*, and *E. seguieriana*, it was aggregated to the areas mentioned above in *A. onobrychis*, *O. petraeus* and *S. nigromaculatus*.”

This is what we now say in the Discussion section with regard to PD complementarity:

“(…) we focused on identifying areas of elevated PD complementarity in the extrazonal steppes that could be prioritized in conservation practice³¹⁻³³ (Figure 2F). We show that PD complementarity is not restricted to or aggregated in single areas; it is, in fact, rather evenly distributed across the extrazonal steppes and exhibits largely species-specific spatial patterns (Figure 4). This again emphasizes the uniqueness and conservation value of extrazonal steppes compared with the zonal steppes as no single region exhibited pervasively elevated phylogenetic diversity (Figure 2F, Figure 4). Given this and the restriction of extrazonal lineages to the steppe outposts, we emphasize that conservation efforts should not concentrate on selected individual areas but rather on the European extrazonal steppes as a whole.”

The following passage in the Discussion has been changed to now read:

“(…) high phylogenetic diversity (Figure 2E) and species-specific, idiosyncratic distribution of PD complementarity (Figure 2F)”

Below are the revised Figure 4 showing the results of PD complementarity calculations and the changed figure caption:

“Figure 4. Distribution of phylogenetic diversity complementarity (PD complementarity) and phylogenetic diversity in six Eurasian steppe species. Grid cells of an equal edge length of one degree encompass sampled populations. PD complementarity values are shown as color of heat. Curves in the insets show the rarefied phylogenetic diversity of extrazonal and zonal populations; the extent of extrazonal and zonal steppes are outlined using the same layers as in Figure 1A (yellow: extrazonal steppes, brown: zonal steppes).”

Reviewer 1, Comment 1.2.

“To conserve the genetic legacy of the Eurasian steppe biome as a whole, each steppe outpost has to be treated as a significant and largely independent entity that evolved in isolation for most of its history.” We ask -□ Where is the evidence for this unique evolution - how many distinct lineages etc are in these bits?

Related to this the claim “To evaluate how much of each species’ diversity was represented by its extrazonal distribution compared with its zonal distribution, phylogenetic diversity was calculated for the respective area62.” - The two totals do not tell us about amount of PD overlap, nor unique PD within each. Such analysis might strengthen the claim that “We found that the evolutionary history of the extrazonal European steppe biota has been largely independent from their zonal relatives, likely since the very onset of the Pleistocene glaciation cycles. Most extrazonal populations belong to lineages entirely absent from the zonal steppes.....” Are there lineages restricted or nearly-restricted to each? It is not revealed in the current analysis. (and Fig 3 is referred to (with unclear use of blue vs red) but seems to cover only extrazonal – how does this inform at all about contrast with zonal?)

Example PD analysis (phylogeography for Europe) can be found in Faith 1992 and later papers.

>>> Response 1.2. Resolved. Thank you for indicating that the distribution of lineages was not made clear enough. Distinct extrazonal lineages restricted to the extrazonal steppes have been detected in all investigated species and are a central finding of this study. We emphasized this, e.g., in the results section titled “Phylogenetic inference exhibited well-supported clades that consisted of exclusively extrazonal lineages in all species (Figure 3)”. However, we agree that this was not evident enough in the earlier version of Figure 3. We have now accounted for this by adding a bar above each STRUCTURE bar plot that indicates the origin of the respective branch/population (extrazonal or zonal) using similar colors as in Figure 1A. To make Figure 3 more intuitively comprehensible, we have also added the layers that delimit the zonal and extrazonal steppes that are used in Figure 1A, again using the corresponding colors.

By implementing PD complementarity analyses and illustrating the distribution of each lineage in Figure 3, we think that it is now evident that unique lineages, endemic to the

extrazonal steppe, are present across all taxa. We think that – by closely following the suggestions provided by Reviewer 1 – the claim that the extrazonal steppes need to be treated as independent and conservation-relevant entities whose biota have evolved in isolation, is now strongly supported and much more evident than in the earlier version of the manuscript.

Below is the revised version of Figure 3 plus the respectively changed caption:

“Figure 3. Patterns of genome-wide Restriction Associated DNA based divergence in six Eurasian steppe species illustrated as maximum likelihood phylogenies (nodes with bootstrap support > 75% are indicated with a black dot) and STRUCTURE⁴⁴ bar plots (colors illustrate

STRUCTURE based gene pools). The maps show the STRUCTURE-based gene pools' geographic distributions. Colored grids above the bar plots indicate the location of the sampled populations in the extrazonal (yellow) or zonal steppes (brown); the extents of extrazonal and zonal steppes are outlined via the layers used in Figure 1A (yellow: extrazonal steppes, brown: zonal steppes; not provided in the small inset showing exclusively zonal localities). Maps in the upper right corner of each panel show potential refugia for cold and warm stages resulting from ecological niche modelling. Projected logistic probabilities of suitability above the species-specific Maximum Training Sensitivity Plus Specificity thresholds are indicated in color."

>>>Additional correction concerning PD accumulation curves:

While revisiting the data to conduct the PD complementarity analysis, we have found that some extrazonal samples were wrongly assigned as stemming from zonal localities in the original PD accumulation curve analysis. This has been corrected. The subsequent calculation of the PD accumulation curve has shown that the extrazonal steppes harbor larger PD than the zonal steppes in the case of *Euphorbia seguieriana*, for which it was the other way round in the paper's earlier version. However, the corrected result corresponds with those for all other taxa now, and the corrected line graphs have been inserted in Figure 4 (see revised Figure 4 below Response 1.1).

The following changes have been made in the text.

Results section:

"The extrazonal steppes harboured substantially higher diversity than the zonal steppes, as shown by comparative rarefaction of phylogenetic diversity (PD, Figure 4)."

Discussion:

"Thus, the extrazonal steppes also harbour a larger intraspecific phylogenetic diversity than the zonal steppes (Figure 4)."

Reviewer 1, Comment 2.

endemism and evol. potential

The CANAPE related analyse of PD endemism do not appear to be informative about conservation value, and do not have any clear link to conservation priority regarding evol potential.

The authors say “Following the logics of modern day conservation policy, in which societal consent and continuous funding are central factors to ensure long-term success of any conservation strategy, we focused on identifying European steppe areas with significantly increased evolutionary potential^{21,34} (Figure 2F-G).

But fig 2F-G does not have any clear link to evol potential criteria,

And re

“the conservation value of zonal versus extrazonal areas using the concept of phylogenetic endemism^{17,18}

“phylogenetic endemism exhibited species-specific spatial patterns (Figure 4). This suggests that, in order to conserve the evolutionary potential^{16,19,21} of the extrazonal steppe biota, Again, I do not think that these indices of phylogenetic endemism can be defended as indicating anything about evol potential.

This conclusion therefore seems unwarranted:

“Altogether, phylogenetic endemism exhibited species-specific spatial patterns (Figure 4). This suggests that, in order to conserve the evolutionary potential^{16,19,21} of the extrazonal steppe biota, conservation efforts should not concentrate on selected individual areas but rather on the European extrazonal steppes as a whole, as there is no single region of pervasively elevated conservation value.”

And this conclusion is misleading “The extrazonal steppes, albeit much smaller than the zonal steppes, were found to harbour both elevated phylogenetic diversity and endemic lineages (Figure 2E-F).”

But the analysis does not explore possible lineages actually restricted (endemic) to extrazonal – a revised analysis could do this.

>>> Response 2. Done. The central message we derived from CANAPE analysis was that it is not sufficient to protect selected parts of the extrazonal steppe because the phylogenetic endemism, and correspondingly, evolutionary potential, is rather evenly distributed than aggregated across species. We agree that CANAPE-derived phylogenetic endemism might not be the most straightforward way to underpin such claims, although the link between evolutionary potential and CANAPE endemism has been emphasized (Gonzalez-Orozco et al. 2016).

Thus, implementation of the concept of PD complementarity (see Response 1.1, Figure 4), as suggested by Reviewer 1, has proven to be a much more elegant approach and well supports our claims. The spatial distribution of PD complementarity supports the conclusion previously drawn from the CANAPE analysis more clearly, that is, when compared across species, phylogenetic diversity as a reflection of evolutionary potential is spread evenly across Europe rather than aggregated in specific regions, and conservation efforts hence should aim at the extrazonal steppe as a whole. We have therefore decided to refrain from using CANAPE.

The following passage has been changed in the Introduction:

“Additionally, we asked if – in case the extrazonal steppes were conservation relevant – conservation efforts should focus on the extrazonal steppes as a whole or just on particular parts. To address this question, we employed the concept of PD complementarity to assess the extent of unique PD of individual parts of the extrazonal steppes compared with the zonal steppes¹⁷ (Figure 2F).”

Accordingly, the following passage describing the CANAPE method has been removed from Material & Methods:

“CANAPE¹⁸ was used to identify cells harbouring significantly shorter or longer branches in relation to the entire phylogeny of each species. This hypothesis test is based on an iterative randomization process in which the tips of the input phylogeny are swapped between grid cells while keeping the richness of each cell constant. In the following two-step categorization, cells with significantly high phylogenetic endemism were identified ($\alpha = 0.05$, one-tailed), and these cells were further categorized into paleo-endemic cells (significantly long branches), neo-endemic cells (significantly short branches), and mixed-endemic cells (significantly long and

short, or rare branches) ($\alpha = 0.05$, two-tailed)¹⁸. All steps involved in the procedure were done in the software Biodiverse (version 2.0) and based on the complete ML phylogenies (see section above)⁶⁴. Finally, the influence of the number of branches on the categorization of a cell was tested by comparing the distribution of branch numbers of endemic versus non-endemic cells in a t-test, which yielded no significant difference at $\alpha = 0.05$.”

Corresponding changes were also made to Figure 2. We have removed CANAPE derived results as criteria in respect to conservation relevance of extrazonal steppes (Figure 2F-G). Instead, we have added PD complementarity as a criterion for conservation relevance (Figure 2F).

Here is the revised version of Figure 2 plus the accordingly changed caption:

“Figure 2. Criteria chosen to evaluate the conservation value of biota of European extrazonal steppes. A, distribution of lineages throughout zonal and extrazonal steppes; populations illustrated as circles, colors represent origin of populations (yellow: extrazonal lineage, brown; zonal origin). B, geographic distribution of observed gene pools and admixture between them.

C, time of lineage divergence. D, geographic position of refugia. E, phylogenetic diversity. F, distribution of phylogenetic diversity complementarity (PD complementarity) across species in the extrazonal steppes: even distribution of PD complementarity across species versus spatial aggregation of PD complementarity across all species in a single area (dark red: large value, light red: small value); the scale indicating conservation value does not apply in this figure panel.”

Reviewer 1, Comment 3.

This diagram and legend is Unclear: "Figure 2. ...A, depth of phylogenetic split between observed clusters

>>> Response 3.1. Changed; thank you for pointing this out. The diagram legend of Figure 2A has been changed to "Distribution of lineages". Also, the maps in Figure 2A have been changed to make them more comprehensible; the figure caption has been adapted accordingly. The revised version of Figure 2, and the altered figure caption is shown in Response 2.

Continuation Reviewer 1, Comment 3.

For fig 3 - Sadly, it is not made clear what red vs blue dots are in this critical fig ...it says "(colors illustrate gene pools)." But "gene pool" is never mentioned in the text of the paper

>>> Response 3.2. Changed. We agree that the color scheme and the term "gene pool" were not elaborated clear enough. By "gene pools", we refer to the gene pools identified by STRUCTURE analysis; this information has now been added to the figure caption. The changed figure caption can be found in Response 1.2.

REVIEWER 2

Reviewer 2, Comment 1. Minor remarks Line 35. Add "." after scale

>>> Response 1. Done.

REVIEWER 3

Reviewer 3, Comment 1. *Introduction, lines 70-74. You have used six species (three plants and three animals) as a representative sample of the steppe biome. The number of species selected is a somewhat arbitrary criteria (to me, enough if they are actually representative of different life histories), but the reasons why these six species and not other ones are selected should be explained here. It is also a pity that the species chosen by you are not very representative of the most peripheral region that contain large patches of extrazonal steppe: the Iberian Peninsula. According to Fig. 1A, at present this peninsula has relatively large stretches of steppe that constitute the westernmost tip of this biome, and thus deserving of being sampled more extensively. However, I think that the study is still providing very valuable results, so this should not be regarded in any way as a “fatal” flaw.*

>>> Response 1. The species were selected based on several criteria. As pointed out by Reviewer 3, the selected taxa indeed represent different life history traits, such as for example: wind pollination (*Stipa capillata*) and insect pollination (*Astragalus onobrychis*, *Euphorbia seguieriana*), short lifespan (*Omocestus petraeus*, *Stenobothrus nigromaculatus*) and long lifespan (all other taxa), epigeous lifestyle (*O. petraeus*, *S. nigromaculatus*) and soil-dwelling lifestyle (*P. taurica*). However, even if we aimed to select taxa representing different life history traits, such traits were not decisive for our selection. Foremost, it was important that the selected taxa are characteristic of large parts of the Eurasian steppe biome. For practical reasons, it was furthermore important to select widespread, abundant, and co-occurring species as all samples needed to be collected in extensive field sampling campaigns. Therefore, also phenology was a factor that needed to be considered, that is, we used strong seasonality as a criterion to exclude taxa, such as geophytes, short-lived therophytes, or arthropods with short lifespans such as solitary bees. We initially aimed to include the orthognath spider *Atypus muralis* to have a wider taxonomic scope but instead included a second grasshopper species as not enough individual-rich *Atypus* populations could be sampled. We further tried to avoid potentially taxonomically difficult species, like for example the widespread steppe-dwelling spider *Eresus niger*, and aimed to avoid heteroploid plants, the intentional exception being *Astragalus onobrychis*. As the reviewer highlights the selected species are not very representative of the Iberian steppes. We did not put much focus on the steppes on the Iberian Peninsula in this study

as they had been classified as a Mediterranean rather than a Central Asian vegetation type (Loidi 2017). Such a classification is strongly supported by the fact that – with the exception of *Stipa capillata* – none of the studied taxa is reported to be widespread on the Iberian Peninsula, while *Astragalus onobrychis* is completely absent there. Furthermore, our preliminary data on *S. capillata* from the Iberian Peninsula showed that Iberian populations might in fact belong to a cryptic species.

The following sentence has been added to the introduction (changes highlighted in blue):

“In particular, we compared climatic niche, phylogenetic history, and evolutionary potential¹⁶ of three flowering plants, which are widespread and abundant elements of the Eurasian steppes (the legume *Astragalus onobrychis*, the spurge *Euphorbia seguieriana*, and the grass *Stipa capillata*) and three arthropods (the grasshoppers *Omocestus petraeus* and *Stenobothrus nigromaculatus*, and the ant *Plagiolepis taurica*) to address two nested research questions.”

The following passages have been added to Material & Methods (changes highlighted in blue):

“Six species were selected, which are characteristic, widespread, abundant, and often co-occurring elements of the Eurasian steppes. In addition, we aimed to avoid taxonomically difficult species as well as heteroploid species, the exception being *A. onobrychis*. Samples of the plant species *A. onobrychis* L. (Fabaceae), *E. seguieriana* Neck. (Euphorbiaceae), and *S. capillata* L. (Poaceae), and the animal species *P. taurica* Santschi, 1920 (Formicidae), *O. petraeus* (Brisout de Barneville, 1856), and *S. nigromaculatus* (Herrich-Schäffer, 1840) (both Acrididae) were collected between 2014 and 2017. All species were sampled across their distribution ranges, with extrazonal occurrences more densely sampled than zonal ones. Of the six study species, only *S. capillata* was suggested to be widespread in the Iberian Peninsula. Therefore, we did not exhaustively sample *S. capillata* outside the Pyrenees, as preliminary data suggested that the Iberian populations belong to another, cryptic species. This divergence is supported by the classification of the Iberian steppes as Mediterranean, instead of Central Asian, vegetation type³⁴. In total, 456 populations from 320 localities were sampled (details are given in Supplementary Table 1). The identification of animals to species level was done using the corresponding keys (grasshoppers³⁵, ants³⁶). Collected specimens were stored in silica gel (plants) or 96% ethanol (animals) for further analyses; herbarium vouchers and animal

specimens were stored at the Departments of Botany (herbarium IB) and of Ecology, University of Innsbruck, respectively.”

Reviewer 3, Comment 2. The sampling of the material for genetic analyses is generally sound, with an adequate coverage of the geographical areas where the species occur (perhaps with the relative exception of *Stipa capillata*; see my comments below) as well as sampling sizes. However, I am very curious why the sample sizes for ENM are so low (even after the 5-km radius removing of occurrences) for plants and animals that are relatively common in the European steppes (especially for the plants). For example, if you have a look to GBIF and focus on Europe, you will see that *Stipa capillata* has almost 4000 occurrences (<https://www.gbif.org/species/4143519>), *Astragalus onobrychis* about 3500 (<https://www.gbif.org/species/5342618>), and *Euphorbia seguieriana* over 8000 (<https://www.gbif.org/species/3066179>). A reason may be that the authors have only used their own records (Supplementary Excel table). For example, *Stipa capillata* seems to have many localities throughout the Iberian Peninsula but only two locations are used. Why the authors have not expanded the ENM occurrences using other sources like biodiversity databases (e.g. GBIF, iNaturalist), articles, books, etc.?

>>> Response 2. We are aware of the wide availability of data from public biodiversity databases, we however decided not to use such data and elaborate our considerations in the following.

Public biodiversity databases unfortunately are prone to contain errors, biases, and data quality issues (Maldonado et al. 2015; Robertson et al. 2016; Troudet et al. 2017). The main errors include geographical errors (i.e. poor quality of geographic positioning) that can be hard to identify, such as, records of a species in its native range that are incorrectly located at the top of a mountain range (Robertson et al. 2016).

The largest problem of biodiversity databases, in our opinion, are erroneous taxonomic identifications. For example, all insect taxa included in this study need to be identified by experts and can be confused with at least one syntopic species (*Plagiolepis taurica* and *P. pygmaea*, *Omocestus petraeus* and *O. haemorrhoidalis* or *O. minutus*, females of *Stenobothrus nigromaculatus* and females of *S. lineatus*, *S. stigmaticus*, *S. eurasius* or *S. fischeri*). While misidentifications might be less problematic in widespread taxa, factors

such as morphological crypsis and unclear distribution ranges have to be considered (e.g. Response 1, *Stipa capillata* on the Iberian Peninsula). In a recent publication, we were able to show that two morphologically hard to distinguish species (*Euphorbia seguieriana* and *E. niciciana*) with parapatric distribution ranges, occupy significantly different ecological niches and, consequently, occupied different cold-stage refugia (Frajman et al. 2019, also cited in the main text).

We were considering to use GBIF data before we had started niche modeling, and we want to point out that the available data were strongly biased towards plants with about 3500 to 8800 occurrences per species, compared with 18 to 1430 occurrences per species for animals. We also found that our own sampling adequately represented the geographic distribution in comparison with the available GBIF records from the extrazonal steppes. *Stipa capillata* occurrences on the Iberian peninsula were not included as our preliminary data from Iberian *S. capillata* suggests cryptic speciation (see also Response 1). Concerning the zonal steppes, no occurrences of the studied insects outside of Europe were available. For these reasons, we opted to restrict our modeling approach on Europe where our own sampling was representative.

For all the above reasons, we were very cautious and rather conservative in selecting occurrences for niche modeling, and in the end used only “our” records for which expert identifications and additional evidence (molecular, morphometric) were available.

Reviewer 3, Comment 4. *Material & Methods, lines 303-304. The number of MCMC is always a controversial question when running STRUCTURE, but most recent works (probably as a consequence of the increase of calculation capacity of modern computers) use at least one million MCMC. An additional suggestion is improving the number of runs.*

>>> **Response 4. Done.** All STRUCTURE runs have been repeated, and the number of MCMC iterations has been increased to 2 millions, using a burnin of 200,000 generations (this change has been highlighted in the corresponding material & methods section). Similar to the previously inferred STRUCTURE results, separation into an optimal number of two clusters was suggested for all studied taxa. While this central finding did not change, minor changes in the admixture pattern in *Astragalus onobrychis* (more admixture on the southern Balkan Peninsula) and *Euphorbia seguieriana* (more

admixture in Pannonia) were found. We emphasize that this finding does not affect the conclusions drawn for this study. Figure 3 has been updated correspondingly, as has been Supplementary Figure 2, which contains the summary statistics for evaluation of the optimal number of STRUCTURE clusters.

Please find the revised versions of Figure 3 and Supplementary Figure 2 as well as the corresponding captions below:

“Figure 3. Patterns of genome-wide Restriction Associated DNA based divergence in six Eurasian steppe species illustrated as maximum likelihood phylogenies (nodes with bootstrap support > 75% are indicated with a black dot) and STRUCTURE⁴⁴ bar plots (colors illustrate STRUCTURE based gene pools). The maps show the STRUCTURE-based gene pools’

geographic distributions. Colored grids in the bar above the bar plots indicate the location of the sampled populations in the extrazonal (yellow) or zonal steppes (brown); the extent of extrazonal and zonal steppes is outlined in the map via layers using the similar color scheme (not in the small inset showing exclusively zonal localities). Maps in the upper right corner of each panel show potential refugia for cold and warm stages resulting from ecological niche modelling. Projected logistic probabilities of suitability above the species-specific Maximum Training Sensitivity Plus Specificity thresholds are indicated in color.”

“Supplementary Figure 2. Summary statistics of STRUCTURE²² analyses of the six analyzed species. Values of Ln probability of the model for each number of groups (K) are plotted against K values in the upper row and delta K value coefficients among runs against K values in the lower row.”

Reviewer 3, Comment 5. *Given that you have included additional information regarding the methods as supplementary material, some methodological questions on ENM should be included here: (1) on the basis of what criteria the 11 variables have been selected from the 19 ones (expert one?); (2) what replication method has been used with Maxent (bootstrap, subsample, cross-validation) and how many replicates have been run; (3) what interpolation method has been used to transform the bioclimatic variables from 2.5 arc-min (the resolution at which these variables are deposited in WorldClim database) to 30 sec. □*

>>> Response 5. Thank you very much for pointing this out! In the following, we reply to

the suggestions (1) to (3) raised by Reviewer 3:

(1) The eleven uncorrelated variables (Pearson's correlation < 0.9) have been selected based on expert opinion. To clarify this, the following passage in the Material & Methods section has been modified (changes marked in blue):

"Pairwise correlation between variables was assessed using ENMTools (version 1.4.4)⁵⁷, and variables with a Pearson's correlation coefficient > 0.9 were removed **based on expert knowledge. Two variables (bio18, bio19) were excluded for technical reasons (Supplementary Methods).** For the final models, eleven variables were used (bio2, bio3, bio4, bio8, bio9, bio10, bio11, bio12, bio15, bio16, bio17; see Supplementary Methods). All ENMs were generated using MAXENT (version 3.3.3.k)⁵⁸."

Additionally, in the Supplementary Methods it has been made clearer now what is meant by technical reasons (changes marked in blue):

"Finally, the resulting surfaces were added to each of the five original bioclim variables for the area encompassing the inner-Alpine dry valleys (while leaving the rest of the layer unaltered). These dry valley-specific variables were used along with the unaltered temperature-related bioclim variables (bio1 – bio11) and the unmodified Precipitation Seasonality (bio15) for all subsequent analyses. **Bio18 and bio19 could not be corrected using the above approach and were therefore excluded as predictors in subsequent analyses.**"

(2) Done. A cross-validation replication regime with a number of replicates equal to the number of species occurrences (k=n) was used.

This is indicated in the Supplemental Material of the original submission (l. 57–61):

“In detail, combinations of regularization parameters ($\beta = 0.5; 1; 1.5; 2; 2.5; 3$) and feature classes (L: linear; H: hinge; LQ: linear-quadratic; and LQH: linear-quadratic-hinge) have been evaluated in regard to model performance using the R package ENMeval version 0.2.2⁸ under a jackknife (k=n) cross validation replication regime.”

In implementing this criticism in the manuscript, we have now made it better visible

where this information can be found, and the revised passage reads:

“All ENMs were generated using MAXENT (version 3.3.3.k)⁵⁸ (model parameters and details on the replication method are described in Supplementary Methods).”

(3) Done. All bioclimatic variables have been obtained in 30 arc-sec resolution as indicated in the main text (including the LGM variables, which were obtained from Schmatz et al. 2015 = Reference 21 in the main text).

The passage in the main text of the original submission reads:

“Bioclim variables for current climatic conditions were obtained from Worldclim v.1.4 (<http://www.worldclim.org/>)⁵⁶ at a resolution of 30 arc-seconds and clipped to the area of study encompassing the European distribution of the six steppe species.”

“The resulting models were projected to conditions of the Last Glacial Maximum, based on MIROC3.2²⁴ and CCSM3²⁴, at 30 arc-seconds spatial resolution.”

Reviewer 3, Comment 6. *Material & Methods, lines 335-337. Your efforts to correct the precipitation variables for mountain areas are commendable. It is a pity that such correction cannot be done for other mountain areas where problems of inaccuracy are probably the same, such as the Pyrenees, the Apennines or the Carpathians.*

>>> Response 6. **Thank you very much for the positive feedback! We have performed those corrections only for the Alps since they represent the topographically and climatically most heterogeneous European mountain range and at the same time the only one with “true” dry valleys.**

Reviewer 3, Comment 7. *Results, lines 137-140. Finding traces of niche divergence between zonal and extrazonal occurrences is a very interesting and significant result, as this is congruent with the genetic data (i.e., evolutionary history of extrazonal steppes independent from zonal ones) as one may expect (see e.g. Xu et al., 2015, Ann Bot. 116: 35–48), and adds more conservation value to the extrazonal steppes. I strongly suggest to explore for this seemingly niche differentiation, by expanding the niche comparative analyses to the other five taxa and, if possible, moving to the E-space. The methodologies developed by several authors in recent*

years are versatile and can be applied to cases with low number of occurrences (but see my comment no. 2), including that of McCormack et al. (2010, *Evolution* 64: 1231–1244) and that of Broennimann et al. (2012, *Global Ecol. Biogeogr.* 21: 481–497).

>>> **Response 7. Done.** Thank you for pointing us towards the possibility of applying the method of McCormack et al. (2010). As suggested by you, we have now explored niche differentiation also for the other species using this method, which is more suitable indeed for low samples sizes. The niche divergence between zonal and extrazonal populations, which had been evident for *S. capillata* through an ENM background test, has now been confirmed for all six species using the McCormack et al. (2010) background test. Due to redundancy of the ENM background test of *S. capillata* with the McCormack et al. (2010) background test (which is available for all six species now), we have decided to replace the former with the latter.

Several modifications of the text were necessary (changes highlighted in blue), and a new Supplementary Table 5 was added to the Supplementary Information.

Results:

“In other words, those zonal localities were predicted as unsuitable because their climatic conditions differed considerably from the extrazonal conditions. This niche difference was confirmed using the background test developed by McCormack et al.¹⁹. The background tests showed that zonal and extrazonal localities of the six species exhibit significant levels of niche divergence (Supplementary Table 5, Supplementary Methods, Supplementary Results).”

Supplementary Methods:

“Extrazonal and zonal sampling localities of steppe species differ in their macro-climatic conditions. This was shown by using extrazonal and zonal occurrences of the six steppe species and by applying the Background Test as described by McCormack et al.¹². This multivariate method does not rely on ENMs but rather compares differences in the environmental background to determine if two sets of localities are more or less similar than expected based on their environmental background^{12,13}. For each species, climatic values of the same bioclim variables as used for ENM were extracted at the species localities as were 1000

random background points using the functionalities of ArcGIS v. 10. 4 (ESRI, Redland, CA). Background area was defined as circular buffer of 100 km in diameter around the occurrence points. PCA were performed and for each species, the first four component scores (PC1–PC4) and the differences for each PC axis between zonal and extrazonal localities were calculated and compared with a null distribution^{12,13} (generated by calculating the difference between background points using a bootstrapping approach and 1000 resamples). The null hypothesis that two sets of occurrences are as similar as expected based on their environmental background is rejected if the observed difference in PC score is lower (= niche conservatism) or higher (= niche divergence) than the 95% confidence limits of the null distribution. The tests were performed using the R package *boot*¹⁴ (version 1.3-23) following the script provided by Johnson et al.¹³.”

Supplementary Results:

“For each species, the first four component scores (PC1–PC4) explained 90–92% of the total variation (Supplementary Table 5). Evidence of niche divergence between zonal and extrazonal localities was detected in all species and in 14 out of 24 niche axes tested. For each species, significant divergence was detected in two to three PC axes explaining a cumulative variance of 21–55 % (Supplementary Table 5). In summary, extrazonal populations occupy areas that are climatically significantly more divergent from those of zonal populations than expected by chance.”

Please find Supplementary Table 5 summarizing the results of background tests below:

Supplementary Table 5. Divergence on niche axes between zonal and extrazonal localities of the six steppe species. Bold values indicate significant niche divergence (D) or conservatism (C) compared with null distribution (in parentheses) based on background divergence between the respective geographic ranges.

Pairwise comparison □(zonal vs. extrazonal)	Niche axes			
	PC1	PC2	PC3	PC4
Astragalus onobrychis	2.63	1.47	0.99	0.90
Null distribution (95% CI)	(2.45, 2.73)	(1.13, 1.41)	(0.22, 0.45)	(0.23, 0.37)
Result	NS	D	D	D
% variance explained	43	25	15	6
Euphorbia seguieriana	1.76	1.57	1.63	0.76
Null distribution (95% CI)	(2.07, 2.40)	(1.71, 1.95)	(0.61, 0.84)	(0.42, 0.55)
Result	C	C	D	D
% variance explained	44	26	16	5
Stipa capillata	2.12	0.92	1.63	0.84
Null distribution (95% CI)	(2.37, 2.68)	(1.11, 1.41)	(0.36, 0.59)	(0.43, 0.57)
Result	C	C	D	D
% variance explained	40	29	15	6
Omocestus petraeus	2.87	1.14	1.52	0.54
Null distribution (95% CI)	(2.64, 2.94)	(1.19, 1.47)	(0.54, 0.74)	(-0.02, 0.11)
Result	NS	C	D	D
% variance explained	44	26	13	7
Plagiolepis taurica	2.80	1.40	0.97	1.14
Null distribution (95% CI)	(2.63, 2.91)	(1.07, 1.36)	(0.37, 0.58)	(0.49, 0.64)
Result	NS	D	D	D
% variance explained	42	27	13	7
Stenobothrus nigromaculatus	3.70	1.20	0.56	0.84
Null distribution (95% CI)	(2.86, 3.15)	(1.53, 1.78)	(0.63, 0.82)	(0.36, 0.49)
Result	D	C	C	D
% variance explained	49	24	13	5

References cited in the Response:

Frajman, B., Závieská, E., Gamisch, A., Moser, T. & Schönswetter, P. Integrating phylogenomics, phylogenetics, morphometrics, relative genome size and ecological niche modelling disentangles the diversification of Eurasian *Euphorbia seguieriana* s. l. (Euphorbiaceae). *Mol. Phylogenet. Evol.* (2018).doi:<https://doi.org/10.1016/j.ympev.2018.10.046>

González-Orozco C.E., Pollock L.J., Thornhill A.H., Mishler B.D., Knerr N., Laffan S.W., Miller J.T., Rosauer D.F., Faith D.P., Nipperess D.A., Kujala H., Linke S., Butt N., Külheim C., Crisp M.D., Gruber B. 2016. Phylogenetic approaches reveal biodiversity threats under climate change. *Nat. Clim. Chang.* 6:1110–1114.

Loidi, J. in *The Vegetation of the Iberian Peninsula. Plant and Vegetation*, vol 12. (ed. Loidi, J.) 513–547 (Springer, 2017).

Maldonado, C., Molina, C. I., Zizka, A., Persson, C., Taylor, C. M., Albán, J., ... & Antonelli, A. (2015). Estimating species diversity and distribution in the era of Big Data: to what extent can we trust public databases?. *Global Ecology and Biogeography*, 24(8), 973-984.

McCormack, J. E., Zellmer, A. J., & Knowles, L. L. (2010). Does niche divergence accompany allopatric divergence in *Aphelocoma* jays as predicted under ecological speciation?: insights from tests with niche models. *Evolution: International Journal of Organic Evolution*, 64(5), 1231-1244.

Robertson, M. P., Visser, V., & Hui, C. (2016). Biogeo: an R package for assessing and improving data quality of occurrence record datasets. *Ecography*, 39(4), 394-401.

Troudet, J., Grandcolas, P., Blin, A., Vignes-Lebbe, R., & Legendre, F. (2017). Taxonomic bias in biodiversity data and societal preferences. *Scientific Reports*, 7(1), 9132.

REVIEWERS' COMMENTS:

Reviewer #1 (Remarks to the Author):

The authors have done a great job addressing my concerns in the review. Perhaps there is one remaining concern – easily fixed.

1) I had noted in my main comment: “This is a good key finding and is documented well in the figs (accum curves), but the case for conservation calls for more information – what PD does the extrazonal have that the zonal does not have? In the Faith 1992 PD study on the phylogeography level, these contributions are PD complementarity values and they are important to the conservation case.”

The authors did an extensive useful analysis adding in PD complementarity values – for individual grid cells within the extrazonal. They describe this well in their revised methods “geographic entities, that is, grid cells of one-degree edge length (i.e. approximately 100 km),

were defined for the extrazonal steppes prior to the analysis. For each of these grid cells, PD complementarity was calculated” and they cover the results in Fig. 4.

But, given that a key conclusion is that the extrazonal has to be treated as a whole, we need a more complete answer to my question “what PD does the extrazonal have that the zonal does not have?” this is the overall complementarity value of the region, not that of individual grid cells.(more details on this can be found in my original review)

It will be easy to add this calculation and discuss.

Emphasis at the same time can be made to the idea that given this complementarity for the extrazonal, one cannot find it all, across all groups, in any one grid cell. A good added sentence in the revision was: “PD complementarity) across species in the extrazonal steppes: even distribution of PD complementarity across species versus spatial aggregation of PD

complementarity across all species in a single area”

2) as an added thought, in my review, I also had noted “the value of PD in conservation could be better highlighted for the reader – e.g. it is used by IPBES and the EDGE program. (see e.g. <https://danielpfaith.wordpress.com/pages/>”

In the revision, the authors now refer to a PD values paper, “Faith, D. P. in The Routledge Handbook of Philosophy of Biodiversity 69–85 (Routledge, 2016). doi:10.4324/9781315530215” and this reference could be used to refer to why we value PD (option value). This will help explain why in fig. 2 PD is included as a criterion.

Dan Faith

Reviewer #3 (Remarks to the Author):

All my questions has been properly answered while all my suggestions has been appropriately addressed. Thus, to me, the mannuscript it is now in Good shape.

Point-by-point replies to the Referee

Referee 1. Comment 1. I had noted in my main comment: “This is a good key finding and is documented well in the figs (accum curves), but the case for conservation calls for more information – what PD does the extrazonal have that the zonal does not have? In the Faith 1992 PD study on the phylogeography level, these contributions are PD complementarity values and they are important to the conservation case.”

The authors did an extensive useful analysis adding in PD complementarity values – for individual grid cells within the extrazonal. They describe this well in their revised methods “geographic entities, that is, grid cells of one-degree edge length (i.e. approximately 100 km), were defined for the extrazonal steppes prior to the analysis. For each of these grid cells, PD complementarity was calculated” and they cover the results in Fig. 4. But, given that a key conclusion is that the extrazonal has to be treated as a whole, we need a more complete answer to my question “what PD does the extrazonal have that the zonal does not have?” this is the overall complementarity value of the region, not that of individual grid cells.(more details on this can be found in my original review)

It will be easy to add this calculation and discuss. Emphasis at the same time can be made to the idea that given this complementarity for the extrazonal, one cannot find it all, across all groups, in any one grid cell. A good added sentence in the revision was: “PD complementarity across species in the extrazonal steppes: even distribution of PD complementarity across species versus spatial aggregation of PD complementarity across all species in a single area”

>>>Response 1. Done. As pointed out by the referee, a comparison of the extrazonal steppes as a whole with the zonal steppes using the concept of phylogenetic diversity (PD) complementarity was still missing from the revised manuscript. Following the reviewer’s suggestions, these values have been calculated and are now provided in a Supplementary Table. The accordingly calculated PD complementarity showed to be larger in the extrazonal steppes compared with the zonal steppes in all six studied species. We thank the reviewer for this useful suggestion. This evidence adds further support to one of our central claims that the extrazonal steppes need to be treated as a whole in future conservation strategies.

The following changes have been made in the Introduction (changes marked in blue):

“To address this question, we employed the concept of PD complementarity to assess the extent of unique PD of the extrazonal steppes as a whole and of individual parts of the extrazonal steppes compared with the zonal steppes¹⁷ (Figure 2F).”

In the Methods section, the following sentence has been added:

“Additionally, PD complementarity for both all extrazonal steppes and all zonal steppes was calculated for each taxon. This was done to compare the amount of unique PD within each area, again using PDA (version 1.0.3)⁶⁴”

In accordance with this finding, the following statement has been added in the results section:

Also, PD complementarity (i.e. the amount of unique PD represented by a subset of a phylogenetic tree that is not present in a reference set) was found to be larger in the extrazonal steppes compared with the zonal steppes in all six taxa (Supplementary Table 5).

We also discuss this additional evidence in the Discussion as follows (changes are marked in blue):

“The extrazonal steppes, albeit much smaller than the zonal steppes, were found to harbour both elevated phylogenetic diversity, a larger amount of unique phylogenetic diversity, and endemic lineages (Figure 2E-F, Supplementary Table 5).”

We show that PD complementarity is generally larger in the extrazonal steppes compared with the zonal steppes and, within the extrazonal steppes, not restricted to or aggregated in single areas; it is, in fact, rather evenly distributed across the extrazonal steppes and exhibits largely species-specific spatial patterns (Figure 4, Supplementary Table 5).

(...) high and at the same time unique phylogenetic diversity (Figure 2E, Supplementary Table 5) and a species-specific, idiosyncratic distribution of this PD complementarity phylogenetic diversity for all six taxa (Figure 2F).

Referee 1. Comment 2. As an added thought, in my review, I also had noted “the value of PD in conservation could be better highlighted for the reader – e.g. it is used by IPBES and the EDGE program. (see e.g. <https://danielpfaith>)”. In the revision, the authors now refer to a PD values paper, “Faith, D. P. in *The Routledge Handbook of Philosophy of Biodiversity* 69–85 (Routledge, 2016). doi:10.4324/9781315530215” and this reference could be used to refer to why we value PD (option value). This will help explain why in fig. 2 PD is included as a criterion.

>>> Response 2. Done. We agree with the referee that some information and additional references concerning the concept of PD and its applications will make the topic more accessible to a reader.

We have added the following sentence to the Results section to offer this general information on PD to a reader prior to presenting the PD results:

“Phylogenetic diversity (PD) is an effective measure to assess conservation value by revealing unknown diversity patterns and unanticipated evolutionary processes irrespective of any taxonomic

classification²²; it is therefore a powerful and widely used measure to inform conservation strategies²³. As such, we interpret PD as a benchmark for diversity and evolutionary processes in the framework of the Eurasian steppes.”